# Latent Wasserstein Adversarial Imitation Learning

**Siqi Yang**    **Kai Yan**    **Alexander G. Schwing**    **Yu-Xiong Wang**
University of Illinois Urbana-Champaign
{siqiyang, kaiyan3, aschwing, yxw}@illinois.edu
https://github.com/JackyYang258/LWAIL

## Abstract

Imitation Learning (IL) enables agents to mimic expert behavior by learning from demonstrations. However, traditional IL methods require large amounts of medium-to-high-quality demonstrations as well as actions of expert demonstrations, both of which are often unavailable. To reduce this need, we propose Latent Wasserstein Adversarial Imitation Learning (LWAIL), a novel adversarial imitation learning framework that focuses on state-only distribution matching. It benefits from the Wasserstein distance computed in a dynamics-aware latent space. This dynamics-aware latent space differs from prior work and is obtained via a pre-training stage, where we train the Intention Conditioned Value Function (ICVF) to capture a dynamics-aware structure of the state space using a small set of randomly generated state-only data. We show that this enhances the policy's understanding of state transitions, enabling the learning process to use only one or a few state-only expert episodes to achieve expert-level performance. Through experiments on multiple MuJoCo environments, we demonstrate that our method outperforms prior Wasserstein-based IL methods and prior adversarial IL methods, achieving better results across various tasks.

## 1 Introduction

As a powerful tool for solving sequential decision-making problems, Reinforcement Learning (RL) has achieved remarkable success in recent years across various fields, such as gaming (Silver et al., 2016) and training large language models (Ramamurthy et al., 2023; Guo et al., 2025). However, RL relies heavily on well-defined reward signals (Jain et al., 2021; Li et al., 2021), which can be difficult to obtain in real-world settings (e.g., robot control (Ibarz et al., 2021) with varied target tasks) or may require careful, environment-specific considerations (Yu et al., 2020).

Imitation Learning (IL) offers a compelling alternative by learning directly from expert demonstrations, thus bypassing the need for explicit reward design. Within IL, a major challenge are the frequently missing expert actions, encouraging adoption of Imitation Learning from Observations (LfO), which uses only sequences of expert states. However, even these state-only demonstrations are often expensive to acquire, making it crucial to develop methods that can learn from a minimal amount of expert data. Adversarial Imitation Learning (AIL) methods (Torabi et al., 2018b; Zhu et al., 2020), which learn by matching the distribution of agent states to that of the expert, are a popular approach to LfO. Many such AIL methods measure the distribution difference by $f$-divergence, such as KL-, $\chi^2$- or JS-divergence. However, $f$-divergences require "distribution coverage", i.e., the distributions in $f$-divergence must be on the same support set to avoid numerical error. This could lead to additional theoretical constraints, especially when an additional dataset of non-expert data is involved; for example, offline LfO methods such as SMODICE (Ma et al., 2022) require the non-expert state distribution to fully cover the expert state distribution, which does not necessarily hold in practice especially when the non-expert data quality is low (e.g., random).

To address these limitations, the Wasserstein distance has gained popularity (Arjovsky et al., 2017; Zhang et al., 2020). However, many prior Wasserstein IL works that employ the Kantorovich-Rubinstein (KR) dual (Zhang et al., 2020; Garg et al., 2021; Sun et al., 2021) overlook an important issue: the distance metric between individual states is rather simplistic. Indeed, to compute the

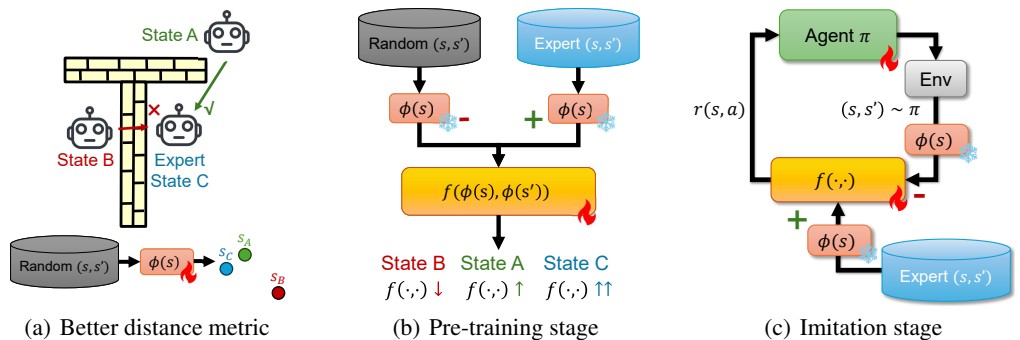

(a) Better distance metric     (b) Pre-training stage     (c) Imitation stage

Figure 1: Illustrating our motivation for a better distance metric and an outline of the algorithm. Panel (a) illustrates a case where the Euclidean distance between states is not a good metric: state B is closer to expert state C, but it is apparently less desirable than more distant state A, as it cannot reach C. To fix, we use random data and ICVF to find a more meaningful embedding space, as shown in the lower half. Panel (b), together with the lower half of panel (a), shows our pre-training stage: we first train ICVF to obtain $\phi(s)$, which serves as a reward for our agent in the online stage shown in panel (c). Fire indicates trainable modules, and snowflakes indicate frozen modules.

Wasserstein distance between distributions of states, a "ground metric" or cost for measuring the distance between any two individual states is first defined. For this, the Euclidean distance is common (Zhang et al., 2020; Garg et al., 2021). However, it fails to capture the environment's dynamics (see Fig. 1(a) for an illustration). This deficiency can severely mislead the learning process. Existing solutions bypass the issue by using the primal Wasserstein distance, which however introduces other complexities and requires surrogates (Yan et al., 2024a; Dadashi et al., 2021). This raises a critical question: *Can we learn a distance metric that encodes the environment's dynamics from a minimum amount of state-only low-quality data*, allowing an agent to learn efficiently from just a few, state-only expert trajectories?

To answer this question, we introduce a novel Wasserstein AIL algorithm that directly addresses this foundational issue. We propose a two-stage process. First, in the pre-training stage, we leverage a small (1% of online rollouts) number of unstructured, low-quality (e.g., random) state-only data to train an Intention-Conditioned Value Function (ICVF) (Ghosh et al., 2023). The resulting embedding captures a rich, dynamics-aware notion of reachability between states. Second, during the imitation stage, we freeze this ICVF embedding and use the Euclidean distance in this new latent space as the cost function within a standard Wasserstein AIL framework. By modifying the agent and discriminator to operate in this learned space, we fix the core issue of prior methods.

We find that this simple approach allows an agent to effectively match the expert's state distribution, enabling highly efficient imitation from minimal expert data. We validate our approach on Umaze and challenging locomotion tasks in the MuJoCo environment from the D4RL benchmark (Fu et al., 2020), and show that the latent space grasps the transition dynamics much better than the vanilla Euclidean distance. In terms of imitation learning, we achieve strong results using only a single trajectory of state-based expert data.

We summarize our contributions as follows: 1) We show that the ICVF latent space captures a dynamics-aware ground metric even if only a small amount of (possibly low-quality) state-only data is available; this is a first direct remedy for the known geometric limitations of the Euclidean distance in prior Wasserstein occupancy matching methods. 2) We propose a simple but effective improvement to current Wasserstein AIL methods achieving expert-level performance with a single state-only expert trajectory; 3) We empirically show that our method outperforms various baselines on multiple testbeds, proving that a better distance metric can greatly benefit AIL.

## 2   PRELIMINARIES

**Markov Decision Process (MDP).** A Markov Decision Process (MDP) is a mathematical framework which describes the interactions of an agent with an environment at discrete time steps. It is

defined by the tuple $(\mathcal{S}, \mathcal{A}, P, R, \gamma)$, where $\mathcal{S}$ represents the state space, and $\mathcal{A}$ denotes the action space. The state transition probability function $P(s'|s, a)$ defines the likelihood of transitioning to a new state $s' \in \mathcal{S}$ after taking an action $a \in \mathcal{A}$ in the current state $s \in \mathcal{S}$. The reward function $R(s, a) \in \mathbb{R}$ specifies the immediate reward received after taking action $a$ in state $s$. $\gamma \in [0, 1)$ is the discount factor, determining the importance of future rewards relative to immediate ones.

At each time step $t$, the agent observes the current state $s_t \in \mathcal{S}$, selects an action $a_t \in \mathcal{A}$, receives a reward $r_t = R(s_t, a_t)$, and transitions to the next state $s_{t+1}$ according to the transition function $P$. A complete running process is called a *trajectory*. The goal of the agent is to learn a policy $\pi : \mathcal{S} \to \mathcal{A}$ that maximizes the expected cumulative discounted reward $G_t = \sum_{k=0}^{\infty} \gamma^k r_{t+k}$. In this paper, we focus on the *state and state-pair occupancy*, which are the visitation frequency of states and state-pairs. Given policy $\pi$, the state occupancy is defined as $d_s^\pi(s) = (1 - \gamma) \sum_{t=0}^{\infty} \gamma^t \Pr(s_t = s)$ and the state-pair occupancy is given by $d_{ss}^\pi(s, s') = (1 - \gamma) \sum_{t=0}^{\infty} \gamma^t \Pr(s_t = s, s_{t+1} = s')$.

**Wasserstein Distance.** Wasserstein distance, also known as Earth Mover's Distance (EMD) (Kantorovich, 1939), is widely used to measure the distance between two probability distributions. For the metric space $(M, c)$ where $M$ is a set and $c : M \times M \to \mathbb{R}$ is a metric, the 1-Wasserstein distance[1] between two distributions $p(x)$ and $q(x)$ on the metric space $(M, c)$ is defined as:

$$\mathcal{W}_1(p, q) = \inf_{\Pi(p,q)} \int_{M \times M} c(x, y) \, d\Pi(x, y). \tag{1}$$

Intuitively, this equation quantifies the optimal way to "move" mass from $p$ to $q$ while minimizing the total movement, as described by the joint distributions $\Pi(p, q)$ with marginals $p$ and $q$. A more popular form adopted by the machine learning community is the Kantorovich-Rubinstein (KR) dual (Kantorovich & Rubinstein, 1958) of the 1-Wasserstein distance, which reads as follows:

$$\mathcal{W}_1(p, q) = \sup_{\|f\|_L \leq 1} \left( \mathbb{E}_{x \sim p}[f(x)] - \mathbb{E}_{x \sim q}[f(x)] \right). \tag{2}$$

Here, $\|f\|_L \leq 1$ restricts function $f$ to be 1-Lipschitz, i.e., for any $x, x'$, $\frac{|f(x) - f(x')|}{c(x, x')} \leq 1$. As the most prominent way to compel Lipschitzness is regularization of the gradient (Gulrajani et al., 2017; Stanczuk et al., 2021) (i.e., $\nabla f(x_0) = \frac{f(x) - f(x_0)}{\|x - x_0\|_2}$ for local $x$), the 1-Lipschitz constraint inherently limits the distance metric $c$ to be Euclidean (Stanczuk et al., 2021), which is often undesirable (Yan et al., 2024a). In this paper, we fix this issue by introducing an ICVF-learned distance metric.

**Intention Conditioned Value Function (ICVF).** ICVF (Ghosh et al., 2023) is an embedding algorithm which learns an augmented value function $V(s, s_+, z)$. It is defined as the unnormalized state occupancy of the future state $s_+ \in \mathcal{S}$ starting from state $s \in \mathcal{S}$ with policy $\pi_z$ that is optimal for reaching some *intention* (i.e., goal state) $z \in \mathcal{S}$. Formally,

$$V(s, s_+, z) = \mathbb{E}_{\pi_z} \left[ \sum_{t=0}^{\infty} \gamma^t p(s_t = s_+ | s_0 = s) \right] = \mathbb{E}_{s_{t+1} \sim P_z(\cdot | s_t)} \left[ \sum_{t=0}^{\infty} \gamma^t \mathbb{I}(s_t = s_+) \mid s_0 = s \right], \tag{3}$$

where $P_z(s_{t+1}|s)$ is the transition probability from $s_t$ to $s_{t+1}$ when acting according to $\pi_z$, which can be seen as a pseudo-policy that takes the next state as "pseudo-action". $\mathbb{I}(s_t = s_+)$ serves a set of pseudo-reward labels: its value is 1 if the condition is true, and 0 otherwise. Thus, $V$ with all different $\pi_z, P_z$ and $s_+$ can be jointly learned from a small, random state-only dataset with offline RL, such as IQL (Kostrikov et al., 2022).

With such a value function learned, the next step is to extract the dynamic-aware embedding. To easily do so, the value function is structured as follows:

$$V_\theta(s, s_+, z) = \phi_\theta(s)^T T_\theta(z) \psi_\theta(s_+). \tag{4}$$

Here, $\phi_\theta(s) \in \mathbb{R}^d$ is the *state representation* that maps a state into a latent space, $T_\theta(z) \in \mathbb{R}^{d \times d}$ is the matrix of *intention*, and $\psi_\theta(s_+) \in \mathbb{R}^d$ is the *outcome representation* (see Appendix C for details). As we later prove in Sec. 3.2, under certain assumptions, the state-pair occupancy of the expert policy is approximately a linear combination of $\phi_\theta(s)$, which justifies our choice of using the ICVF embedding. This design is also empirically validated in Sec. 3.2 and Sec. 4.

---

[1]Unless otherwise specified, we will discuss 1-Wasserstein distance in this paper.

## 3 LATENT WASSERSTEIN ADVERSARIAL IMITATION LEARNING (LWAIL)

This section is organized as follows: In Sec. 3.1 we first define our goal and frame it using a Wasserstein adversarial state occupancy matching objective with KR duality. We then point out its inherent shortcomings and propose the ICVF-trained latent space metric as a solution in Sec. 3.2. Finally, we introduce our algorithm in Sec. 3.3. See Fig. 1 for an overview of our work.

### 3.1 WASSERSTEIN ADVERSARIAL STATE OCCUPANCY MATCHING

Our goal is to learn a policy $\pi$ using three sources of information: a few-shot, state-only expert dataset $E$, another small dataset $I$ with state-only *random* transitions $(s, s')$ (either given or collected with an untrained policy), and online interactions. Inspired by recent state occupancy matching works (Kostrikov et al., 2020; Garg et al., 2021; Ma et al., 2022; Kim et al., 2022a), here, we minimize the 1-Wasserstein distance between state-pair occupancy distributions of the policy $\pi$, i.e., $d_{ss}^\pi(s, s')$, and of the empirical policy of the expert, i.e., $d_{ss}^E(s, s')$. Formally, we address

$$\min_\pi \mathcal{W}_1(d_{ss}^\pi(s, s'), d_{ss}^E(s, s')), \tag{5}$$

where $s$ and $s'$ are adjacent states in a trajectory. Note, while many occupancy matching works, such as SMODICE (Ma et al., 2022) and LobsDICE (Kim et al., 2022a), use $f$-divergences, we opt to use the 1-Wasserstein distance because it provides a smoother measure and leverages the underlying geometric property of the state space, unlike $f$-divergences.

However, the Wasserstein distance itself is hard to compute as it is inherently a constrained linear programming problem (see Eq. (1)), which is difficult to solve via gradient descent. While there exist workarounds such as convex regularizers (Yan et al., 2024a), surrogates (Dadashi et al., 2021), and direct matching of trajectories (Luo et al., 2023; Bobrin et al., 2024), here, we choose the widely adopted KR dual (Kantorovich & Rubinstein, 1958) as our objective. Combined with policy optimization, this results in the final objective

$$\min_\pi \max_{\|f\|_L \leq 1} \left( \mathbb{E}_{(s,s') \sim d_{ss}^\pi}[f(s, s')] - \mathbb{E}_{(s,s') \sim d_{ss}^E}[f(s, s')] \right), \tag{6}$$

where the 1-Lipschitz constraint can be encouraged by prominent methods such as gradient regularization (Gulrajani et al., 2017). With constraint 'addressed,' Eq. (6) is a bi-level optimization and can be optimized iteratively by any RL algorithm. Specifically, since $d_{ss}^E$ is independent of $\pi$, the objective for finding policy $\pi$ is $\max_\pi \mathbb{E}_{(s,s') \sim d_{ss}^\pi}[-f(s, s')]$. This can be optimized with any RL algorithm using reward $r(s, a) = \mathbb{E}_{s' \sim P(s'|s,a)}[-f(s, s')]$. From an adversarial imitation learning perspective, $f(s, s')$ can be interpreted as a discriminator that outputs a high score $f(s, s')$ for expert state pairs and a low score $f(s, s')$ for non-expert ones.

### 3.2 DYNAMICS-AWARE DISTANCE METRIC WITH ICVF EMBEDDING SPACE

While the objective in Eq. (6) already provides a viable solution, it has a subtle limitation: As mentioned in Sec. 2, the metric $c(s, s')$ is limited to be Euclidean in practice due to the Lipschitz constraint $\frac{\|f(s)-f(s')\|}{c(s,s')} \leq 1$. However, as illustrated in Fig. 1, a Euclidean distance in the raw state space often fails to capture the true relation between states. This subtle reliance on the Euclidean distance is often ignored by prior KR duality-based imitation learning methods, e.g., IQlearn (Garg et al., 2021) and WDAIL (Zhang et al., 2020), but crucial for performance (Yan et al., 2024b).

To address this issue, we aim to learn a latent state representation, in which the Euclidean distance serves as a more effective metric that permits capturing the environment's dynamics and relationship between states from a *small amount of randomly collected, unlabeled data, even without access to ground-truth actions and rewards*. To achieve this goal, we benefit from the ICVF (Ghosh et al., 2023) framework. We found that using as the cost $c(s, s') = \|\phi_\theta(s) - \phi_\theta(s')\|_2$ the root mean squared difference between state representations in latent space, i.e., $\phi_\theta(s)$, is not only empirically effective as validated in Sec. 4, but it also theoretically benefits learning. More specifically, we have:

**Theorem 3.1.** *In a near-deterministic MDP with $\gamma < 1$, For any converged policy $\pi_z$ learned in ICVF with goal $z$, there exists a vector $\eta$ such that, for any adjacent $(s, s') \in I$, $d_{ss}^{\pi_z}(s, s') \approx \eta^T \phi_\theta(s)$, i.e., the state-pair occupancy is approximately a linear combination of $\phi$.*

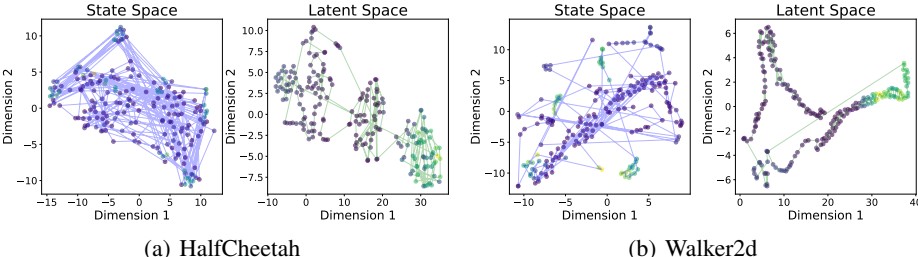

(a) HalfCheetah           (b) Walker2d

Figure 2: t-SNE visualization of the same trajectory in the original state space and the embedding latent space. The color of the points represents the ground-truth reward of the state (greener is higher). States connected by lines are adjacent in the trajectory. We observe that an ICVF-trained embedding provides a much more dynamics-aware metric than the Euclidean distance. For Hopper and Ant environment embeddings, see Appendix F.4.2.

We defer the proof to Appendix D. Intuitively, the idea is that the value function $V(s, s', z)$, our optimization target $d_{ss}^\pi(s, s')$, and the expert state-pair occupancy $d_{ss}^E(s, s')$, all model the probability of visiting $s'$ after $s$ with policy $\pi_z$. As they share the same underlying linear structure of $\phi_\theta$, which captures the environment's dynamics, learning in the space of $\phi_\theta$ can reduce the learning difficulty by aligning via ICVF the computation structure of Wasserstein optimization and transition dynamics. The benefits of an alignment have also been observed in other contexts (Xu et al., 2020).

To further show the effectiveness of the distance metric $c(s, s') = \|\phi_\theta(s) - \phi_\theta(s')\|_2$, we provide a t-SNE (Van der Maaten & Hinton, 2008) visualization of the same trajectory in both the raw state space and the latent space in Fig. 2 using environments we evaluated in Sec. 4.2. The result shows that the latent space better captures the dynamic relationship between states. This finding highlights that the Euclidean distance in the latent space is a more suitable metric for Wasserstein distance-based state matching. We also provide ablations in Sec. 4.4 and Appendix F.

### 3.3 LWAIL

We now introduce our proposed method, LWAIL, which consists of two stages: pre-training and imitation. We will also provide pseudo-code to summarize our approach.

**Pre-training.** The pre-training stage consists of three steps. First, collect random transition data from the environment by using a randomly initialized policy. This can be skipped if a random dataset is available, e.g., in a setting following Yue et al. (2024). Second, train the value function via the loss given in Eq. (11) using IQL, and retrieve the projection function $\phi$ from the value function formulated in Eq. (4). Third, train $f(\cdot, \cdot)$ using the following objective with frozen latent variable mapping $\phi$ and frozen, untrained, random policy $\pi$:

$$\max_{\|f\|_L \leq 1} \left( \mathbb{E}_{(s,s') \sim d_{ss}^\pi}[f(\phi(s), \phi(s'))] - \mathbb{E}_{(s,s') \sim d_{ss}^E}[f(\phi(s), \phi(s'))] \right). \tag{7}$$

Here, $f$ serves as the discriminator (from an adversarial learning perspective), the reward function for the policy in later imitation (from an RL perspective), and the KR dual function (from a Wasserstein perspective). To encourage the Lipschitz constraint, $f$ is trained with a gradient penalty, following WGAN-GP (Gulrajani et al., 2017).

**Imitation.** In the online imitation learning stage, we again freeze the learned embedding $\phi$ and replace $s$ and $s'$ with their latent space representations, $\phi(s)$ and $\phi(s')$. Then the imitation learning problem in Eq. (6) can be addressed via

$$\min_\pi \max_{\|f\|_L \leq 1} \left( \mathbb{E}_{(s,s') \sim d_{ss}^\pi}[f(\phi(s), \phi(s'))] - \mathbb{E}_{(s,s') \sim d_{ss}^E}[f(\phi(s), \phi(s'))] \right). \tag{8}$$

Following the standard off-policy approach, the agent interacts with the environment to gather data and iteratively updates the value function and policy. Once the policy has collected a batch of trajectories, we update the discriminator network $f$ based on Eq. (8). Following Sec. 3.1, using an adversarial learning framework, we then use $f$ to generate rewards for the downstream reinforcement learning algorithm, for which we employ TD3 (Fujimoto et al., 2018), a robust, off-policy

---

**Algorithm 1** LWAIL

---

**Require:** State-only expert dataset $E$, state-action random dataset $I$ (optional), initial policy $\pi$, discriminator $f$, replay buffer $\mathcal{B}$, update frequency $m$

    **Pretrain:**
1: Collect transitions into buffer $\mathcal{B}$ with random actions or use random dataset provided
2: Use ICVF to pre-train the representation network $\phi$ (Eq. (11))
3: Pre-train $f$ with initial policy $\pi$ (inner level of Eq. (8))
4: **Imitation:**
5: **while** $t \leq T$ **do**
6:     Collect transitions $(s, a, s', \text{done})$ using $\pi$
7:     Get pseudo-reward $r_p = \sigma(-f(\phi(s), \phi(s')))$
8:     Add $(s, a, s', r_p, \text{done})$ to replay buffer $\mathcal{B}$
9:     **if** $t \mod m == 0$ **then**
10:       Update $f$ (inner level of Eq. (8))
11:     **end if**
12:     Sample mini-batch of $N$ transitions from $\mathcal{B}$ to perform TD3 update
13: **end while**

---

reinforcement learning method selected due to its stability and effectiveness. Slightly different from Sec. 3.1 however, the reward for the TD3 policy $\pi$ is defined as $r(s, s') = \sigma(-f(\phi(s), \phi(s')))$. $\sigma$ is the sigmoid function that normalizes the reward to the range $[0, 1]$, stabilizing the downstream RL algorithm. $-f(\phi(s), \phi(s'))$ is a 1-sample estimation of $\mathbb{E}_{s' \sim P(s'|s,a)}[-f(\phi(s), \phi(s'))]$ for transition $(s, a, s')$, following Kim et al. (Kim et al., 2022a). $f$ and $\pi$ are then iteratively updated until the policy $\pi$ is properly trained. The entire procedure of our method is summarized in Alg. 1.

## 4 EXPERIMENTS

In this section, we assess the efficacy of LWAIL across multiple benchmark tasks. Specifically, we study the following questions: 1) How is our assigned reward $f(\phi(s), \phi(s'))$ different from the ground-truth reward? 2) Can our algorithm work well on complicated continuous control environments? 3) How much do the ICVF embedding LWAIL contribute to its performance, and is LWAIL robust to environment noise? We will answer 1) in Sec. 4.1, 2) in Sec. 4.2, and 3) in Sec. 4.4.

### 4.1 SIMPLE ENVIRONMENT ON MAZE2D

We first evaluate LWAIL in the Umaze environment, which offers intuitive visualizations of how LWAIL effectively learns reward representations for downstream tasks.

**Experimental and Dataset Setup.** We use the maze2d-umaze-v0 environment from D4RL (Fu et al., 2020), where a 2D point mass moves from a random start to a specific target at coordinates (1,1). Observations consist of $(x, y)$ positions and velocities; actions are linear forces in $x, y$. The sparse reward is 1 when the goal is reached (distance $< 0.5$m). The ICVF model is trained on the D4RL random dataset, followed by Wasserstein learning with Euclidean and latent-space distances.

**Results.** After convergence, the reward map is compared in Fig. 3 (b) and (c). The state used to calculate rewards is the position with zero velocity. The environment walls are shown in orange. These results indicate that the ICVF-learned metric captures trajectory dynamics, enhancing reward feedback during online inverse RL exploration. The reward curve in Fig. 3 (d) further shows that our method converges effectively on Maze2d, surpassing TD3 with ground-truth sparse rewards.

### 4.2 MUJOCO ENVIRONMENT

**Baselines.** We test a variety of baselines in this section, which can be categorized into four types: 1) *classic imitation methods*, including GAIL (Ho & Ermon, 2016), AIRL (Fu et al., 2018) and the plain Behavior Cloning (BC); 2) *Wasserstein-based imitation methods*, including PWIL (Dadashi et al., 2021), WDAIL (Zhang et al., 2020) and IQlearn (Garg et al., 2021); 3) *LfO methods*, including BCO (Torabi et al., 2018a), GAIfO (Torabi et al., 2018b), DIFO (Huang et al., 2024), LS-IQ (Al-

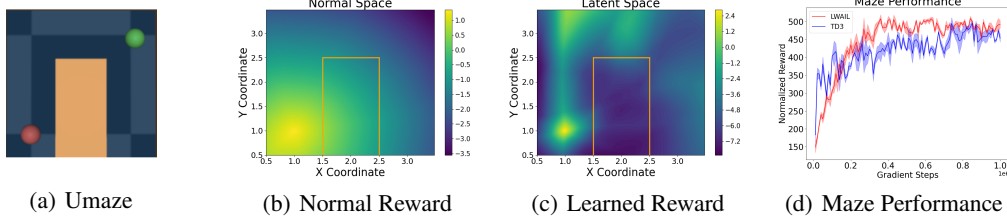

| (a) Umaze | (b) Normal Reward | (c) Learned Reward | (d) Maze Performance |

Figure 3: Results on Maze2D. Panel (a) shows the environment setup with the start (green) and the goal (red). Panel (b) and (c) illustrate the reward distribution without and with ICVF embedding (orange represents the wall). Panel (d) shows the learning curves: LWAIL vs. TD3 with ground-truth rewards. We note that the reward is more dynamics-aware with LWAIL.

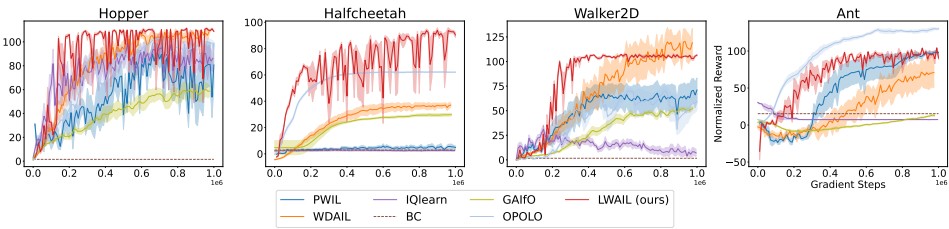

Figure 4: Performance on the MuJoCo environments. Overall, our method consistently delivers strong performance across all tasks. For LWAIL, the number of gradient steps matches the number of online interaction samples. For clarity, we present training curves for main baselines here; please refer to Fig. 7 for the complete version.

Hafez et al., 2023), DiffAIL (Wang et al., 2024), OPOLO (Zhu et al., 2020) and DACfO (LfO variant of DAC (Kostrikov et al., 2019); serves as a baseline for OPOLO (Zhu et al., 2020)); 4) *offline to online imitation learning*, which includes OLLIE (Yue et al., 2024).[2] Some methods, such as GAIL and WDAIL require expert action. For these methods, we report the results with extra access to the expert actions. We report the mean and standard deviation from 5 independent runs with different seeds. The performance is measured by the D4RL-defined normalized reward (higher is better).

**Experimental and Dataset Setup.** We evaluate our method on four standard MuJoCo (Todorov et al., 2012) environments: Hopper, HalfCheetah, Walker2D, and Ant. The expert data for both our method and the baselines consists of a single trajectory from the D4RL expert dataset. The corresponding normalized expert scores are 113.23, 88.42, 106.84, and 116.97, respectively. For ICVF pre-training, we use 10K state pairs collected from random rollouts, which is merely $1\%$ of our online data. All baselines, including ours, are trained with a single state-only trajectory as expert data. We report the normalized average reward over 10 evaluation trajectories, providing both the mean and standard deviation (higher values indicate better performance). Each method is trained with 1M online samples. Additional hyperparameters are provided in Appendix E.3.

**Results.** Fig. 4 shows the reward curves of each method on MuJoCo environments, while Tab. 1 summarizes the final results. Both the figure and the table show that our method achieves compelling results and convergence compared to the baselines across tasks. BCO, DACFO, OPOLO, and PWIL perform well in some environments, while other methods struggle.

### 4.3 NAVIGATION ENVIRONMENT

To further evaluate our method in a setting similar to that discussed in Fig. 1, we test on the maze2d-medium, maze2d-large, and antmaze-umaze-v2 navigation tasks. For maze2d-medium and maze2d-large, to increase the difficulty and investigate the agent's generalization capabilities, we inject Gaussian noise into the agent's initial state, with the noise level being the standard deviation of

---

[2]We are unable to run the github version of OLLIE due to non-trivial typos in their code, and we are not able to reproduce LS-IQ results, likely due to a version mismatch for unspecified packages. For both methods, we directly report the final numbers with 1 expert trajectory from their paper instead.

Table 1: Performance comparison on MuJoCo environments. The first part of the table lists methods that require extra access to expert actions; the second part is for LfO methods. '*' represents results reported from the original paper, as we are unable to reproduce the results on our machine. Results are averaged over 50 trajectories. It is apparent that our method outperforms most baselines, even those with access to expert actions. DIFO is the most competitive expert state-only baseline.

|  | Hopper | HalfCheetah | Walker2D | Ant | Average |
|---|---|---|---|---|---|
| BC | $1.51 \pm 0.61$ | $1.92 \pm 0.80$ | $2.06 \pm 0.99$ | $12.74 \pm 2.34$ | 4.56 |
| GAIL | $7.78 \pm 2.13$ | $-0.33 \pm 0.60$ | $2.14 \pm 1.03$ | $-1.98 \pm 4.41$ | 1.90 |
| AIRL | $1.16 \pm 0.43$ | $6.02 \pm 3.59$ | $0.83 \pm 0.95$ | $3.30 \pm 11.39$ | 2.83 |
| WDAIL | $107.72 \pm 4.70$ | $38.30 \pm 1.09$ | $126.07 \pm 19.36$ | $87.59 \pm 13.12$ | 89.92 |
| DiffAIL | $95.52 \pm 9.94$ | $33.36 \pm 28.14$ | $\mathbf{133.90 \pm 1.41}$ | $80.56 \pm 13.09$ | 85.84 |
| OLLIE | $71.10 \pm 3.5*$ | $35.50 \pm 4.0*$ | $59.80 \pm 8.5*$ | $57.10 \pm 7.0*$ | $55.87*$ |
| PWIL | $78.93 \pm 39.00$ | $20.81 \pm 33.21$ | $84.01 \pm 26.53$ | $105.62 \pm 2.36$ | 72.34 |
| IQlearn | $86.24 \pm 21.92$ | $2.51 \pm 1.05$ | $7.07 \pm 7.23$ | $7.39 \pm 0.13$ | 25.80 |
| BCO | $21.31 \pm 4.06$ | $4.08 \pm 1.72$ | $0.88 \pm 0.85$ | $25.33 \pm 0.87$ | 12.90 |
| DACfO | $109.46 \pm 0.39$ | $61.52 \pm 0.76$ | $45.28 \pm 37.26$ | $113.40 \pm 10.20$ | 82.41 |
| GAIfO | $58.74 \pm 9.07$ | $29.79 \pm 2.12$ | $52.73 \pm 4.16$ | $12.99 \pm 2.77$ | 38.56 |
| OPOLO | $99.24 \pm 5.49$ | $58.98 \pm 7.46$ | $37.07 \pm 12.67$ | $129.46 \pm 3.64$ | 81.19 |
| DIFO | $98.50 \pm 12.60$ | $78.62 \pm 18.68$ | $99.43 \pm 11.72$ | $93.47 \pm 8.91$ | 92.51 |
| LS-IQ | $76.84 \pm 4.41*$ | $41.64 \pm 3.64*$ | $102.02 \pm 2.54*$ | $\mathbf{132.28 \pm 3.29*}$ | $88.20*$ |
| LWAIL (ours) | $\mathbf{108.84 \pm 0.79}$ | $\mathbf{90.40 \pm 3.71}$ | $\mathbf{106.51 \pm 1.14}$ | $90.53 \pm 3.26$ | **99.07** |

the Gaussian distribution. This perturbation forces the agent to start from unseen observations that deviate from the expert demonstrations.

**Maze2d.** The quantitative results summarized in Tab. 2 demonstrate that LWAIL significantly outperforms the baseline with diverse initial state. As the noise level increases, LWAIL without ICVF suffers a catastrophic performance drop, failing to navigate successfully. In contrast, LWAIL with ICVF embedding maintains consistent high performance across all noise levels. This suggests that the ICVF embeddings in LWAIL successfully capture the environment's dynamics, allowing the agent to handle unseen observations and recover from unfamiliar states.

Table 2: Performance on maze2d-medium and maze2d-large with different levels of perturbed initial states. We report the normalized returns across different noise levels.

| Maze Type | Noise Level | LWAIL | No Embedding |
|---|---|---|---|
| Maze2d-medium | 0.0 | $144.59 \pm 0.87$ | $117.91 \pm 0.30$ |
|  | 0.2 | $143.42 \pm 0.91$ | $-17.00 \pm 0.01$ |
|  | 0.5 | $135.12 \pm 3.12$ | $-16.98 \pm 0.02$ |
| Maze2d-large | 0.0 | $156.68 \pm 0.60$ | $148.25 \pm 6.41$ |
|  | 0.2 | $157.04 \pm 0.71$ | $-11.11 \pm 0.04$ |
|  | 0.5 | $157.06 \pm 0.81$ | $-11.13 \pm 0.02$ |

**Antmaze.** Tab. 3 shows the performance on antmaze, for which we use WDAIL, IQ-Learn, and SMODICE (Ma et al., 2022) as the baselines. The result shows that LWAIL with ICVF works comparably well with the baselines, and again illustrates the importance of ICVF embedding. Note, different from SMODICE, which uses different hyperparameters such as divergences (KL vs. $\chi^2$), our method uses the same hyperparameter for all environments.

Table 3: Performance on antmaze-umaze-v2.

| LWAIL | No Embedding | WDAIL | IQ-Learn | SMODICE |
|---|---|---|---|---|
| $34.76 \pm 7.21$ | $0.03 \pm 0.01$ | $31.03 \pm 5.62$ | $0.00 \pm 0.00$ | $44.44 \pm 7.31$ |

Table 4: Ablation of different embedding methods with LWAIL. The results show that ICVF embeddings outperform other contrastive learning-based embeddings.

|  | Hopper | HalfCheetah | Walker2D | Ant | Average |
|---|---|---|---|---|---|
| LWAIL | $108.84 \pm 0.79$ | $90.40 \pm 3.71$ | $106.51 \pm 1.14$ | $90.53 \pm 3.26$ | **99.07** |
| PW-DICE | $110.60 \pm 0.77$ | $46.07 \pm 27.95$ | $106.63 \pm 1.03$ | $85.36 \pm 8.12$ | 87.16 |
| CURL | $105.70 \pm 1.22$ | $87.62 \pm 5.10$ | $102.97 \pm 4.19$ | $52.03 \pm 8.33$ | 87.08 |
| No Embedding | $108.34 \pm 3.42$ | $85.98 \pm 3.42$ | $62.39 \pm 20.43$ | $40.72 \pm 18.95$ | 74.36 |

Table 5: LWAIL with and without environment noise on MuJoCo environments. The result shows that LWAIL is robust to transition noise in the environment.

|  | Hopper | HalfCheetah | Walker2D | Ant | Average |
|---|---|---|---|---|---|
| LWAIL | $110.52 \pm 1.06$ | $86.71 \pm 5.67$ | $105.30 \pm 2.33$ | $80.56 \pm 13.09$ | 95.77 |
| LWAIL with noise 0.1 | $110.25 \pm 1.78$ | $82.04 \pm 6.68$ | $104.98 \pm 1.44$ | $79.95 \pm 12.08$ | 94.31 |
| LWAIL with noise 0.2 | $108.27 \pm 2.38$ | $79.29 \pm 5.21$ | $104.61 \pm 2.34$ | $77.10 \pm 12.08$ | 92.32 |
| LWAIL with noise 0.5 | $107.93 \pm 2.38$ | $48.39 \pm 3.95$ | $41.28 \pm 29.66$ | $-12.94 \pm 9.12$ | 46.17 |

## 4.4 ABLATION STUDY

Due to the page limit, we only present part of the ablations. See Appendix F for more ablations on the dataset, algorithm components, baselines, visualizations, and results on more environments.

**How well does our embedding work?** To further verify the effectiveness of the ICVF embedding beyond theoretical insights, we compare our method with ICVF embeddings to use of other embeddings. We identify two contrastive learning-based baselines that are most suitable for our scenario: CURL (Laskin et al., 2020) and PW-DICE (Yan et al., 2024a). Both methods use InfoNCE (Oord et al., 2018) as their contrastive loss for better state embeddings. Their difference: 1) CURL updates embeddings with an auxiliary loss during online training, while PW-DICE updates embeddings before all other training; 2) CURL compares the current state with different noises added as positive contrast examples, while PW-DICE uses the next states as positive contrast samples. Tab. 4 summarizes the results. The result shows that 1) state embeddings generally aid learning, and thus a good state embedding is necessary; and 2) our proposed method works best.

**How robust is LWAIL with respect to environment noise?** One limitation of our Thm. 3.1 is that it assumes the environment to be near-deterministic. To empirically validate the robustness of our method with noisy transitions, we inject Gaussian noise with varying standard deviations into the actions $a \in [-1, 1]^n$ performed in the MuJoCo environments. This noise introduces controlled randomness, emulating real-world uncertainty and variability in system responses. Other experimental settings follow the main experiments. The results are listed in Tab. 5. The results clearly show that our method is robust to stochastic environments.

**Applying ICVF embedding on baselines.** We also show that our proposed solution outperforms existing methods with ICVF embedding, both Wasserstein-based (IQlearn, WDAIL) and $f$-divergence based. The results are summarized in Tab. 6. We find that 1) our method outperforms prior methods with ICVF embedding, and 2) ICVF best suits our AIL framework with TD3 as the downstream RL algorithm (see Tab. 13 for our method with other downstream RL algorithms).

Table 6: ICVF with other methods. Our method far outperforms baselines with ICVF embeddings.

|  | Hopper | HalfCheetah | Walker2D | Ant | Average |
|---|---|---|---|---|---|
| LWAIL | $108.84 \pm 0.79$ | $90.40 \pm 3.71$ | $106.51 \pm 1.14$ | $90.53 \pm 3.26$ | **99.07** |
| WDAIL+ICVF | $110.02 \pm 0.53$ | $30.07 \pm 2.32$ | $68.68 \pm 9.16$ | $3.42 \pm 1.01$ | 53.04 |
| IQlearn+ICVF | $29.80 \pm 10.12$ | $3.82 \pm 0.98$ | $6.54 \pm 1.23$ | $8.91 \pm 0.45$ | 12.27 |
| GAIL+ICVF | $8.96 \pm 2.09$ | $0.12 \pm 0.40$ | $3.98 \pm 1.41$ | $-3.09 \pm 0.85$ | 2.49 |

## 5    RELATED WORK

Due to the space limit in the main paper, we only discuss three areas of the literature that are most related to our work. See Appendix B for an extended discussion of related work.

**Wasserstein-based imitation learning.** The Wasserstein distance (Kantorovich, 1939) is widely adopted in IL/RL (Xiao et al., 2019; Agarwal et al., 2021; Fickinger et al., 2022). It provides a geometry-aware measure between policy occupancies with more informative learning signals (Arjovsky et al., 2017). Among different forms of the Wasserstein distance, the primal form (Dadashi et al., 2021; Luo et al., 2023; Yan et al., 2024a; Bobrin et al., 2024) and the Rubinstein-Kantorovich dual (Kantorovich & Rubinstein, 1958; Zhang et al., 2020; Garg et al., 2021; Sun et al., 2021) are most prominent. The former allows for larger freedom in its underlying metric, but requires a regularizer (Yan et al., 2024a), surrogates (Dadashi et al., 2021), or a direct match between trajectories (Luo et al., 2023; Bobrin et al., 2024). The latter is easier to optimize with gradient descent, but its metric is limited to Euclidean, which is often suboptimal (Stanczuk et al., 2021; Yan et al., 2024a). Our work chooses the latter but overcomes its shortcomings. Among all these works, IQ-learn (Garg et al., 2021) is most similar to ours. Our objective in Eq. (6) is a special case of IQ-learn with Wasserstein distance. However, three key differences exist: 1) IQ-learn uses SAC (Haarnoja et al., 2018) instead of TD3; 2) IQ-learn focuses on $\chi^2$-divergence in the online setting, which was found to be less effective in several prior works (Ma et al., 2022; Yan et al., 2024a); 3) We point out and overcome the metric limitation by adopting ICVF, which is not considered in IQ-learn.

**Offline-to-online IL.** While offline IL (Zolna et al., 2020; Ma et al., 2022; Kim et al., 2022a) and online IL (Ho & Ermon, 2016; Fu et al., 2018) are both well-studied areas, offline-to-online IL is relatively under-explored, especially when compared with offline-to-online RL (Schmitt et al., 2018; Kostrikov et al., 2022; Zhang et al., 2023a) which combines the best of offline RL (high data efficiency) and online RL (active data collection). While there are some works using offline data to aid online imitation (Watson et al., 2024) by building dynamic models (Chang et al., 2021; Yue et al., 2023) or aligning discriminator and policy (Yue et al., 2024), they differ in two aspects from our proposed LWAIL: 1) their solution requires medium-to-high quality offline data and does not work well with random offline data, with which our ICVF-learned metric works well; 2) they require state-action pairs for expert demonstrations, while our method only requires expert states.

**State embedding.** Many works have explored the possibility of learning a good state space embedding that better captures the dynamics of the environment and boosts RL performance (Zhang et al., 2021; Ghosh et al., 2023; Modi et al., 2024). These works can be roughly categorized into two groups: 1) the 'theoretical group', which focuses on state equivalence (also known as "bisimulation") (Zhang et al., 2021; Kemertas & Aumentado-Armstrong, 2021; Le Lan et al., 2021) and the low-rank property of the MDP (Agarwal et al., 2020; Uehara et al., 2022; Modi et al., 2024); and 2) the 'empirical group' often tested on visual RL with high-dimensional input (Anand et al., 2019; Laskin et al., 2020; Yarats et al., 2022; Giammarino et al., 2023)), which focuses on representation learning (Ha & Schmidhuber, 2018; Hafner et al., 2023; Bruce et al., 2024), autoencoder methods (Senthilnath et al., 2024), and contrastive learning (Sermanet et al., 2017; Anand et al., 2019; Laskin et al., 2020). The recently proposed ICVF (Ghosh et al., 2023) studies an empirical, intention-based method for state embedding computation. It was shown to be effective in downstream tasks (Ghosh et al., 2023; Bobrin et al., 2024). Our work is the first to leverage ICVF state embeddings to overcome the metric limitation of the KR duality of the Wasserstein distance.

## 6    CONCLUSION

We propose a novel adversarial imitation learning approach for state distribution matching with Wasserstein distance. Unlike prior methods that rely on Euclidean distance metrics, we optimize this distance metric by leveraging an embedding learned by the Intention Conditioned Value Function (ICVF), which captures environmental dynamics. This allows us to better align the expert-agent state distributions, even with sparse state-only demonstrations. Through theoretical analysis and multiple experiments, we demonstrate that the ICVF-learned distance metric outperforms baselines, enabling more efficient and accurate imitation from limited expert data with only one expert trajectory. We believe our work provides a new direction for improving state-only imitation learning by using the Wasserstein distance while addressing the limitations of traditional distance metrics.

## ACKNOWLEDGMENTS

This work was supported in part by NSF under Grants 2008387, 2045586, 2106825, and 2519216, the DARPA Young Faculty Award, the ONR Grant N00014-26-1-2099, the NIFA Award 2020-67021-32799, the Amazon-Illinois Center on AI for Interactive Conversational Experiences, the Toyota Research Institute, and the IBM-Illinois Discovery Accelerator Institute. This work used computational resources, including the NCSA Delta and DeltaAI supercomputers through allocations CIS230012, CIS230013, and CIS240419 from the Advanced Cyberinfrastructure Coordination Ecosystem: Services & Support (ACCESS) program, as well as the TACC Frontera supercomputer and Amazon Web Services (AWS) through the National Artificial Intelligence Research Resource (NAIRR) Pilot.

## ETHICS STATEMENT

The proposed imitation learning method enables more efficient policy learning from minimal expert demonstrations (even state-only) and random data, potentially democratizing access to RL applications in robotics and assistive technologies while reducing reliance on costly expert annotations. It is important, however, to utilize our work adequately in real-life applications to avoid potential misuse (e.g. military purposes) and job loss.

## REPRODUCIBILITY STATEMENT

We include the procedure of our algorithm in Alg. 1. For environments used in our experiments, we list their details in Sec. 4.1 (for Maze2d) and Appendix E.1 (for MuJoCo); for datasets in our experiment, we list their statistics in Appendix E.2; for the hyperparameters of our method, we list them in Tab. 8 in Appendix E.3; for implementation of the baselines, their related repositories and licenses, we summarize them in Appendix E.4. Finally, we state our computational resource consumption in Appendix H. We published our code at https://github.com/JackyYang258/LWAIL.

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

APPENDIX: LATENT WASSERSTEIN ADVERSARIAL IMITATION LEARNING

Our appendix is organized as follows. We first discuss limitations of our work in Sec. A, and more areas of related work in Sec. B. In Sec. C, we discuss extended preliminaries of our work, including TD3 and a more detailed explanation of Eq. (11) and ICVF. In Sec. D, we provide a mathematical proof for Thm. 3.1. In Sec. E, we provide the details of the environment in our experiments (Sec. E.1), the dataset used in our experiments (Sec. E.2), the hyperparameters we used for our method (Sec. E.3), and the details for our baselines (Sec. E.4). We provide a variety of ablations in Sec. F. In Sec. G, we summarize the notation used in our paper. Finally, in Sec. H, we state the computational resources used for running our experiments.

In particular, we provide a detailed analysis of our method in Sec. F, which includes the following content:

- **Dataset ablations:** How will the performance change if size (Sec. F.2.1) and quality (Sec. F.2.2) of the offline state-only dataset $I$ change? How will the performance change if the size of the expert dataset $E$ changes (Sec. F.2.3), and how does LWAIL work with incomplete expert trajectories (Sec. F.2.4)?

- **Algorithm component ablations:** What happens if we use other RL algorithms (e.g., PPO (Schulman et al., 2017) or DDPG (Lillicrap et al., 2016)) in the imitation stage (Sec. F.3.1), and is the Sigmoid reward mapping that we use in LWAIL always beneficial for RL on MuJoCo environments (Sec. F.3.2)?

- **Detailed performance analysis:** How robust is our method with respect to different hyperparameters (Sec. F.4.1)? What are the t-SNE visualizations of the trajectories in the embedding space for the Ant and Hopper environment (Sec. F.4.2)? How does our pseudo-reward look like during training (Sec. F.4.3), and how does it perform compared to a ground-truth reward signal (Sec. F.4.4)?

- **Baseline analysis:** How do the baselines perform when they are trained for a longer time (Sec. F.5.1)?

- **More environments:** How does LWAIL perform beyond MuJoCo environments (Sec. F.6.1), and can LWAIL learn from expert trajectories with mismatched dynamics (Sec. F.6.2)?

## A  LIMITATIONS

Similar to other prior AIL methods such as WDAIL (Zhang et al., 2020), our pipeline requires an iterative update of the actor-critic agent and the discriminator during online training. The update frequency needs to be balanced during training. Also, testing our method on more complicated environments, such as image-based ones, is interesting future research.

## B  EXTENDED RELATED WORK

**Imitation by occupancy matching / adversarial training.** GAIL (Ho & Ermon, 2016) is one of the first works to study adversarial imitation learning. A bi-level optimization of "RL over inverse RL" is considered, which corresponds to state-action occupancy matching between an expert and the learner's policy. Follow-up works (Fu et al., 2018; Kostrikov et al., 2019; Torabi et al., 2018b; Zhu et al., 2020) have further explored the adversarial training paradigm, jointly training 1) a discriminator to distinguish policy occupancy from expert occupancy; and 2) the policy fitting expert demonstrations. While some occupancy matching works like DIstribution Corrected Estimation (DICE) (Ma et al., 2022; Kim et al., 2022a; Yan et al., 2024a) are trained using a single-level optimization instead of an adversarial setup, they essentially derive a closed-form solution for the policy given the discriminator. Most works in this field, however, focus on $f$-divergence (especially KL (Zhu et al., 2020) or $\chi^2$ (Ma et al., 2022)) minimization, making them not only unaware of the underlying geometry of the states which leads to less informative learning signals, but also requires them to consider the same support set for the expert and learner occupancy distributions, which could lead to additional theoretical constraints and numerical issues (Ma et al., 2022). In contrast, our work studies the Wasserstein distance and overcomes initial limitations.

*Remark* B.1. Note, the key difference between DICE methods and our Wasserstein AIL framework is that DICE methods are generally not compatible with Wasserstein distance. DICE methods usually use $f$-divergences, especially KL and $\chi^2$-divergences because the Bellman constraint is a linear one; to turn the optimization into an unconstrained one, the primal objective must be non-linear. Unfortunately, Wasserstein distance is a linear objective, hence incompatible with DICE. While there are some exceptions, they must add an additional non-linear regularizer to bypass this issue, such as a state-action KL divergence between learner and non-expert data (Yan et al., 2024a) or an entropy regularizer (Sun et al., 2021). In contrast, our current Wasserstein AIL framework does not exhibit this issue.

*Remark* B.2. Theoretically, with infinite data, perfect convergence and while ignoring regularizers (e.g., offline IL methods like LobsDICE (Kim et al., 2022a) use a pessimistic KL-divergence term with suboptimal dataset occupancy distribution, which deviates from the optimal solution), all transition-distribution matching methods optimizing $(s, s')$, $(s, a)$ or $(s, a, s')$ can encode optimal paths regardless of the underlying distance metric. However, this idealization does not remove the practical need for a geometry-aware embedding in real policy learning. Two reasons are central:

- **Neural networks struggle when semantically different states are numerically close:** If two dynamically distinct states are nearly identical in raw Euclidean space, the value function cannot reliably separate them. A good embedding makes these differences linearly separable and eases optimization substantially.

- **Wasserstein AIL fundamentally depends on the quality of the underlying metric:** Unlike f-divergences, Wasserstein distances are defined directly through the metric structure. A poor metric yields a discontinuous, highly non-smooth loss landscape. Enforcing Lipschitzness with non-Euclidean metrics is difficult in practice; embedding-based Euclidean metrics provide a practical and effective solution.

Thus, even though minimizing transition distributions is theoretically sufficient asymptotically, in practice, a good state embedding is crucial for **learnability, stability, and sample efficiency** under Wasserstein AIL.

**Imitation from observation.** Imitation (Learning) from Observation (LfO) aims to retrieve an expert policy without labeled actions. This is particularly interesting in robotics, where the expert action can be either inapplicable during cross-embodiment imitation (Sermanet et al., 2017) or unavailable when imitating from videos (Pari et al., 2022). The three primary strategies of LfO can be categorized as follows: 1) minimizing an occupancy divergence through DICE methods (Zhu et al., 2020; Lee et al., 2021; Ma et al., 2022; Kim et al., 2022a;b; Yan et al., 2024a) or iterative inverse-RL updates (Torabi et al., 2018b; Xu & Denil, 2019; Zolna et al., 2020); 2) predicting missing actions through inverse dynamics modeling (Torabi et al., 2018a; Kumar et al., 2019); and 3) similarity-based reward assignment (Sermanet et al., 2017; Chen et al., 2019; Wu et al., 2019). Our work belongs to the first category, and adopts the Wasserstein distance as the measure between occupancies, which improves results over prior works.

**Model-based Imitation Learning.** While ICVF implicitly encodes environment dynamics into its embedding space, some imitation learning work takes a more explicit, model-based way to build a dynamic model or world model (Yue et al., 2023). For example, RMBIL (Lin et al., 2021) uses a Neural ODE (Chen et al., 2018) to model state transition, while CMIL (Kolev et al., 2024) uses DreamerV2 (Hafner et al., 2021); many other studies (Chang et al., 2021; Jiang et al., 2020) simply utilize normal neural networks as inverse dynamic models. With such dynamic models, a wide range of options have been explored, including recovering expert action (Torabi et al., 2018a), alignment to new dynamics (Jiang et al., 2020), planning in advance (Yin et al., 2022; Li et al., 2025; Englert et al., 2013; Wu et al., 2020), generating data from simulated rollouts (Chang et al., 2021; Zhang et al., 2023b; Kidambi et al., 2021; Rafailov et al., 2021), and making the environment differentiable (Baram et al., 2016). Among these more explicit model-based methods, ICVF embedding is special as it does not require any action; while many aforementioned methods do not need expert action (Torabi et al., 2018a; Kidambi et al., 2021), they still need non-expert actions to build the dynamic model.

## C   EXTENDED PRELIMINARIES

**TD3.** Twin Delayed Deep Deterministic Policy Gradient (TD3) (Fujimoto et al., 2018) extends the Deep Deterministic Policy Gradient (DDPG) (Lillicrap et al., 2016) algorithm, designed to mitigate the overestimation bias commonly found in Q-learning. TD3 introduces three key modifications: 1) *Clipped Double Q-learning*: TD3 maintains two Q-networks, $Q_{\theta_1}$ and $Q_{\theta_2}$ parameterized by $\theta_1$ and $\theta_2$ respectively, and uses the smaller of the two as the critic loss to reduce overestimation (Lee & Lee, 2023). The clipped value stabilizes training and results. 2) *Delayed Policy Updates*: To further stabilize learning, the policy is updated less frequently than the critic, reducing the chance of policy updates based on inaccurate Q-values. 3) *Target Policy Smoothing*: To address overfitting to deterministic policies, when calculating the target $y$ for the critic loss, a Gaussian noise $\epsilon$ with variance $\sigma^2 > 0$ is clipped with a threshold $c_0 > 0$ before being added to the target action $a'$. This regularizes the policy, making it more robust to small state changes. TD3 improves upon DDPG and works well, particularly in high-dimensional continuous action spaces. In this paper, we adopt TD3 for the downstream RL component of our method.

**Extended Introduction of Intention Conditioned Value Function (ICVF).** Intuitively, $V(s, s_+, z)$ is designed to evaluate the likelihood of the following question: *How likely am I to see $s_+$ in the future trajectory if I act with the intention of reaching $z$ from state $s$?* The learning of ICVF is similar to other value-learning algorithms. For policy $\pi(\cdot|s, s_+)$, ICVF satisfies the following Bellman equation:

$$V^\pi(s, s_+, z) = \mathbb{E}_{z \sim \pi(\cdot|s, s_+)} \left[ \mathbb{I}(s = s_+) + \gamma \mathbb{E}_{s' \sim P_z(\cdot|s_t)} \left[ V^\pi(s', s_+, z) \right] \right]. \tag{9}$$

Here, $(s, s')$ is a transition and $P_z(s_{t+1}|s)$ is the transition probability from $s_t$ to $s_{t+1}$ when acting according to intent $z \in \mathcal{S}$.[3] Here, $z$ serves as a pseudo-action label and $\mathbb{I}(s = s_+)$ serves as a pseudo-reward label. Thus, the original reward or action are not needed in ICVF training.

The original paper adopts implicit Q-learning (IQL) for ICVF learning. In one update batch, we sample transition $(s, s')$, potential future outcome $s_+$, and intent $z$. Similar to the original IQL (Kostrikov et al., 2022), we update the critic with asymmetric critic losses to avoid out-of-distribution overestimation. To do this, we adjust the weights of the critic loss based on the positivity of the *advantage*. Note, as we care about whether the transition $(s, s')$ corresponds to acting with intention $z$, our goal $s_+$ is equal to $z$.[4] The advantage $A$ is defined as:

$$A = \mathbb{I}(s = s_+) + \gamma V_\theta(s', z, z) - V_\theta(s, z, z). \tag{10}$$

Following that, the critic loss is defined as:

$$\mathcal{L}(V_\theta) = \mathbb{E}_{(s,s'),z,s_+} \left[ |\alpha - \mathbb{I}(A < 0)| (V_\theta(s, s_+, z) - \mathbb{I}(s = s_+) - \gamma V_\theta^{\text{old}}(s', s_+, z))^2 \right], \tag{11}$$

where $z \in \mathcal{S}$ and $s_+ \in \mathcal{S}$ are sampled from random states in the dataset, from random future states in the trajectory, or as the current state with different probability, respectively. Note, as we want to penalize overestimation, the coefficient for overestimation $|\alpha - 0| = \alpha$ should be greater than underestimation $|\alpha - 1| = 1 - \alpha$, and thus $\alpha > 0.5$. In LWAIL, we use $\alpha = 0.9$ following the original ICVF paper.

*Remark* C.1. The principle of ICVF embedding is related to *bisimulation metrics* (Castro, 2020). The **bisimulation metric** trains an encoder such that in its embedding space, the distance between the embedding approximately equals the bisimulation metric, which is "reward difference + Wasserstein distance of transition distribution in the embedding space." For example, the "distance between the embedding" can be approximated by another neural network, and the Wasserstein distance is bypassed by the deterministic property of MDP (Castro, 2020). Alternatively, the "distance between the embedding" can be Manhattan distance with 2-Wasserstein distance for a convenient closed form (Zhang et al., 2021). Meanwhile, **ICVF embedding** trains a value function $V(s, s_+, z)$ conditioned on intention (goal state) $z$ and reward function being the visit probability of the state $s_+$. $V(s, s_+, z)$ is designed to be a product of the embedding and other components. They are similar as they:

---

[3]Theoretically, $z$ can also be task-specific labels, e.g., indices for different tasks, but ICVF is mainly motivated by goal-based RL. Thus, in both the main part of ICVF and in our work, we have $z \in \mathcal{S}$.

[4]Intuitively, we use "the best actions with the highest estimated value" of the next step in traditional Bellman operators; here, apparently the intention of reaching $z$ is the best choice if the target future state is $z$.

- Both use Wasserstein distance (ICVF itself does not use Wasserstein distance, but LWAIL does);

- Both address the issue described in Fig. 1(a), which describes the similarity of two states regardless of their numerical values. Bisimulation does so by considering embedded state similarity within one step reach (propagated to trajectories during learning), while ICVF does so by discrimination in different values for the value function with the same intention $z$ and the same reward based on $s_+$.

- Both produce representations that can generalize to unseen reward functions. ICVF does so by carefully designing the value function architecture, such that all downstream value functions with state-based rewards are linear representations of the embedding; bisimulation does so by encoding all the causal ancestors of the reward under certain conditions (see Thm. 4 in Zhang et al. (2021) for details).

The key difference is: *bisimulation is action-focused, while ICVF is state-focused.* This difference leads to different advantages of the two embeddings: ICVF can work with only the state available, but the notion of "intention" assumes goal-based downstream rewards for generalization guarantee; bisimulation metric requires actions for training, and is designed for denser reward functions (since the metric directly compares reward). The ability of ICVF utilizing low-quality state-only data which is otherwise unusable aligns with our motivation.

# D  MATHEMATICAL PROOFS

Here we provide the proof of Thm. 3.1, which we restate for readability:

**Theorem D.1.** *In a near-deterministic MDP with $\gamma < 1$, For any converged policy $\pi_z$ learned in ICVF with goal $z$, there exists a vector $\eta$ such that, for any adjacent $(s, s') \in I$, $d_{ss}^{\pi_z}(s, s') \approx \eta^T \phi_\theta(s)$, i.e., the state-pair occupancy is approximately a linear combination of $\phi$.*

*Proof.* We first consider the trivial case: $z$ is unreachable from $s$. In this case, we have $V(s, s_+, z) = 0$ for any $s_+$, which is apparently a linear combination of $\phi$ with coefficient 0.

Next, consider the definition of the ICVF value function $V^\pi(s, s_+, z) \in \mathbb{R}$. For goal $z$, current state $s$ and some possible future state $s_+$,

$$V^\pi(s, s_+, z) = \sum_{i=1}^{\infty} \gamma^{i-1} p(s_i = s_+ | s_0 = s), \tag{12}$$

where $i$ denotes the future steps. Meanwhile, the state and state pair occupancy $d^\pi(s)$, $d^\pi(s, s_+)$ are defined as

$$d_s^\pi(s) = (1 - \gamma) \sum_{t=0}^{\infty} \gamma^t p(s_t = s),$$

$$d_{ss}^\pi(s, s_+) = (1 - \gamma) \sum_{t=0}^{\infty} \gamma^t p(s_{t+1} = s_+, s_t = s). \tag{13}$$

Therefore for any adjacent $(s, s_+) \in I$,

$$\begin{aligned}
&d_s^\pi(s) \cdot V^\pi(s, s_+, z) \\
=&(1 - \gamma) \left( \sum_{t=0}^{\infty} \gamma^t p(s_t = s) \right) \left( \sum_{i=1}^{\infty} \gamma^{i-1} p(s_{i+t} = s_+ | s_t = s) \right) \\
=&(1 - \gamma) \left( \sum_{t=0}^{\infty} \gamma^t p(s_t = s) \right) \left( p(s_{t+1} = s_+ | s_t = s) + \sum_{i=2}^{\infty} \gamma^{i-1} p(s_{i+t} = s_+ | s_t = s) \right) \\
=&d_{ss}^\pi(s, s_+) + (1 - \gamma) \left[ \sum_{t=0}^{\infty} \gamma^t p(s_t = s) \sum_{i=2}^{\infty} \gamma^{i-1} p(s_{i+t} = s_+ | s_t = s) \right].
\end{aligned} \tag{14}$$

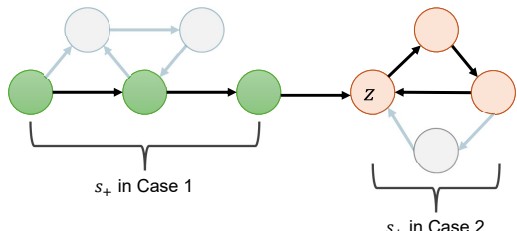

Figure 5: An illustration of "case 1" (green states) and "case 2" (orange states) in our proof, and the $z$ is the intention (i.e. goal). The transparent states are longer paths, which are unlikely be visited with a converged policy and $\gamma < 1$ in a near-deterministic MDP.

We now consider two different cases of $s_+$. They correspond to the two cases illustrated in Fig. 5. We will prove that either the second term is negligible (case 1), or it is a multiple of $d_{ss}^\pi(s, s_+)$ (case 2).

**Case 1: $s_+$ is on the path towards reaching $z$.**

Since $s_+$ is steadily reachable from $s$ in a near-deterministic environment, visiting $s'$ directly from $s$ instead of later returning from other states to $s'$ shortens the trajectory towards $z$. It should thus be preferred by any converged policy $\pi_z$ with $\gamma < 1$. Thus, we have

$$p(s_{i+1} = s_+|s_t = s) \gg p(s_{i+k} = s_+|s_t = s), k \geq 2 \text{ and } (s, s_+) \in I, \tag{15}$$

which means the $(1 - \gamma) \left[\sum_{t=0}^\infty \gamma^t p(s_t = s) \sum_{i=2}^\infty \gamma^{i-1} p(s_{i+t} = s_+|s_t = s)\right]$ term *is negligible*. In this case, $d_{ss}^\pi(s, s_+) \approx d_s^\pi(s) \cdot V(s, s_+, z) = d_s^\pi(s) \cdot \phi_\theta(s)^T T_\theta(z) \psi_\theta(s_+)$, and we have $\eta = d_s^\pi(s) \cdot T_\theta(z) \psi_\theta(s_+)$.

**Case 2: $s_+$ is on the path after reaching $z$.**

In this case, $\pi_z$ should "loop back" to $z$ as soon as the MDP allows. As the MDP is near-deterministic, suppose the smallest state cycle looping back towards $z$ is $T$. Without loss of generality, we assume $z = s_+$; for other states in the loop, a factor of $\gamma^i$ applies if the agent is still $i$ steps away from $s_+$ in the loop.

Then, we have

$$p(s_{t+i} = s_+ \mid s_t = s) \approx \begin{cases} 0, & i \geq 2, \ (i-1) \bmod T \neq 0, \\ p(s_{t+1} = s_+ \mid s_t = s), & i \geq 2, \ (i-1) \bmod T = 0. \end{cases} \tag{16}$$

Thus, we have

$$\begin{aligned}
&d_s^\pi \cdot V^\pi(s, s_+, z) \\
=&(1 - \gamma) \left(\sum_{t=0}^\infty \gamma^t p(s_t = s)\right) \left[p(s_{t+1} = s_+|s_t = s) + \sum_{i=2}^\infty \gamma^{i-1} p(s_{i+t} = s_+|s_t = s)\right] \\
=&\frac{1}{1 - \gamma^T} d^\pi(s, s_+),
\end{aligned} \tag{17}$$

and thus similar to case 1, $\eta = (1 - \gamma^T) d^\pi(s) \cdot T_\theta(z) \psi_\theta(s_+)$.

$\square$

*Remark* D.2. An example of case 1 in the proof is a navigation task; in a navigation task, the agent moves towards its goal without hovering over any intermediate state $s'$. An example of case 2 is keeping a pendulum inverted after reaching its balance (Foundation, 2025), which requires the agent to stay at the current state. Note, all MuJoCo environments tested in Sec. 4 satisfy the assumption in Thm. 3.1, as the noise on the initial states is mild while the transition functions are deterministic. We also find LWAIL to be empirically robust when the assumption does not hold, e.g., with noise in transition (Sec. 4.4).

# E    EXPERIMENTAL DETAILS

## E.1    ENVIRONMENTS

We use five MuJoCo (Todorov et al., 2012) and D4RL (Fu et al., 2020) environments: Maze2d, Hopper, HalfCheetah, Walker2D and Ant. The environment specifications for maze2d are provided in Sec. 4.1. In this section, we will briefly introduce the other MuJoCo environments. Fig. 6 provides an illustration of those environments.

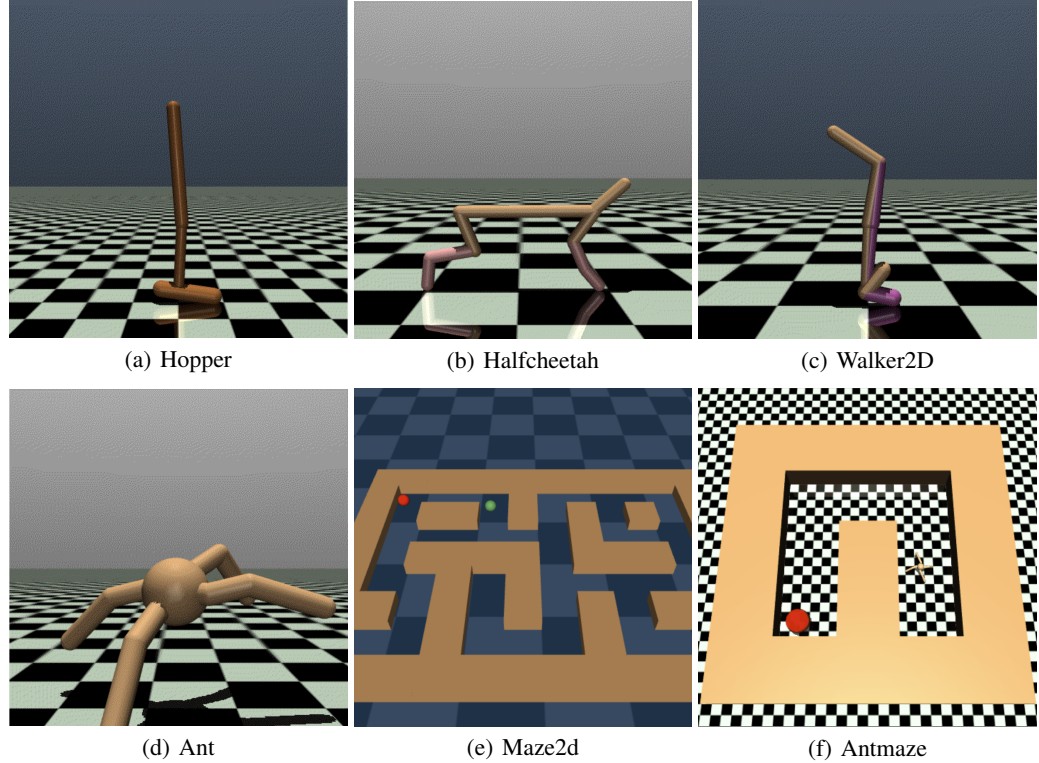

|  (a) Hopper | (b) Halfcheetah | (c) Walker2D |

|  (d) Ant | (e) Maze2d | (f) Antmaze |

Figure 6: Illustration of the MuJoCo (Todorov et al., 2012) environments we test in Sec. 4.2 and Sec. 4.3.

1. **Hopper.** The hopper environment (as well as the other three environments) is a locomotion task. In Hopper, the agent needs to control a single-legged robot leaping forward in a 2D space with $x$- and $z$-axis. The 11-dimensional state space encompasses joint angles and velocities of the robot, while the 3-dimensional action space corresponds to torques applied on each joint.

2. **Halfcheetah.** In the Halfcheetah environment, the agent needs to control a cheetah-shaped robot to sprint forward. It also operates in a 2D space with $x$- and $z$-axis, but has a 17-dimensional state representing joint positions and velocities, and a 6-dimensional action space that modulates joint torques.

3. **Walker2D.** As implied by its name, in Walker2D, the agent needs to control a 8-DoF bipedal robot to walk in the two dimensional space. It has a 27-dimensional state space and an 8-dimensional action space.

4. **Ant.** Different from the previous three environments, the Ant environment is a 3D setting where the agent navigates a four-legged robotic ant moving towards a particular direction. The state is represented by 111 dimensions, including joint coordinates and velocities, while the action space has 8 dimensions.

5. **Maze2d.** In maze2d, the agent controls a 2-DoF, force-actuated ball in a maze to reach the target goal. The action space is 2-dimensional, which represents the force applied on each

dimension; the observation space is 4-dimensional, describing the velocity and location of the agent.

6. **Antmaze.** Antmaze is a more difficult version of maze2d, where the agent needs to navigate through the maze, not with a ball, but with a robotic ant similar to that in the ant environment. The state space is 29-dimensional and the action space is 8-dimensional.

### E.2 DATASETS

For expert datasets of the MuJoCo and navigation environments, we use 1 trajectory from the D4RL expert dataset (with Apache-2.0 license) as $E$, which has 1000 steps, and randomly sample trajectories with a total of 10K transitions from the D4RL random dataset as $I$. Some baselines, such as PWIL (Dadashi et al., 2021) employ a *subsampling* hyperparameter, which creates a low-data training task by taking only one state/state-action pair from every 20 steps of the expert demonstration. For fairness, we set all baselines' subsampling factors to be 1, i.e., no subsampling except for Sec. F.2.4.

Table 7: The basic statistics of the random datasets from D4RL (Fu et al., 2020) applied in our MuJoCo experiments (dataset for navigation tasks are random explorations without reward). We use the first 10K transitions in each dataset for ICVF training.

It is apparent that all these data are of very low quality compared to an expert, yet our ICVF-learned metric still works well.

| Dataset | Normalized Reward (Expert is 100) |
|---|---|
| Hopper-random-v2 | $1.19 \pm 1.16$ |
| HalfCheetah-random-v2 | $0.07 \pm 2.90$ |
| Walker2d-random-v2 | $0.01 \pm 0.09$ |
| Ant-random-v2 | $6.36 \pm 10.07$ |

### E.3 HYPERPARAMETERS

Tab. 8 summarizes the hyperparameters for our method. We use the same settings for all environments, and keep hyperparameters identical to TD3 (Fujimoto et al., 2018) and ICVF (Ghosh et al., 2023) whenever possible.

### E.4 BASELINES

We use several different Github repositories for our baselines. We use default settings of those repos, except for the number of expert trajectories (which is set to 1) and the subsampling factor (see Appendix E.2). We always follow the original hyperparameters in their paper if possible; otherwise, we use hyperparameters from similar experiments. Below are the repos we used in our experiments for each baseline:

- *BC (Ross et al., 2011), GAIL (Ho & Ermon, 2016), AIRL (Fu et al., 2018):* We use the *imitation* (Gleave et al., 2022) library, which provides clean implementations of several imitation learning algorithms and has a MIT license.

- *OPOLO (Zhu et al., 2020), DACfO (Kostrikov et al., 2019), BCO (Torabi et al., 2018a), GAIfO (Torabi et al., 2018b):* We use OPOLO's official code (https://github.com/illidanlab/opolo-code), where DACfO, BCO and GAIfO are integrated as baselines, which does not have a license.

- *OLLIE (Yue et al., 2024):* We tried to use the official code but it can't be executed due to non-trivial typos. Thus we use their reported numbers on D4RL random dataset instead.

- *PWIL:* We use another widely adopted imitation learning repository (Arulkumaran & Ogawa Lillrank, 2023) (https://github.com/Kaixhin/imitation-learning), which has an MIT license.

Table 8: Summary of the hyperparameters of LWAIL.

| Type | Hyperparameter | Value | Note |
|------|----------------|-------|------|
| ICVF. | Network Size of $\phi$ | [256, 256] | same as original paper |
| Disc. | Network Size | [64, 64] | |
| | Activation Function | ReLU | |
| | Learning Rate | 0.001 | |
| | Update Epoch | 40 steps | |
| | Update interval | 4000 | |
| | Batch Size | 4000 | |
| | Optimizer | Adam | |
| | Gradient Penalty coefficient | 10 | |
| Actor | Network Size | [256, 256] | |
| | Activation Function | ReLU | |
| | Learning Rate | 0.0003 | |
| | Training length | 1M steps | |
| | Batch Size | 256 | |
| | Optimizer | Adam | |
| Critic | Network Size | [256, 256] | |
| | Activation Function | ReLU | |
| | Learning Rate | 0.001 | |
| | Training Length | 1M steps | |
| | Batch Size | 256 | |
| | Optimizer | Adam | |
| | $\gamma$ | 0.99 | discount factor |

- *WDAIL:* We use their official code (https://github.com/mingzhangPHD/Adversarial-Imitation-Learning/tree/master), which does not have a license.

- *IQlearn:* We use their official code (https://github.com/Div99/IQ-Learn/tree/main) with a research-only license.

- *DIFO:* We use their official code (https://github.com/NTURobotLearningLab/DIFO/tree/main) with a research-only license.

- *DiffAIL:* We use their official code (https://github.com/ML-Group-SDU/DiffAIL) which does not have a license. As the training process of DiffAIL is highly unstable (especially on the walker environment), we select the runs that reach 1M step if possible.

- *LS-IQ:* We tried to use their official code (https://github.com/robfiras/ls-iq) with a MIT license, but the result cannot be reproduced on their dataset on our machine, likely due to an environment version mismatch. Thus, we use their reported numbers with 1 expert trajectory.

- *SMODICE:* We use their official code (https://github.com/JasonMa2016/SMODICE) which does not have a license.

# F  MORE ABLATIONS

In this section, we provide additional ablation results of our method. We report normalized reward (higher is better) for all results.

## F.1  COMPLETE TRAINING CURVES

Tab. 7 presents the full training curves corresponding to the main results.

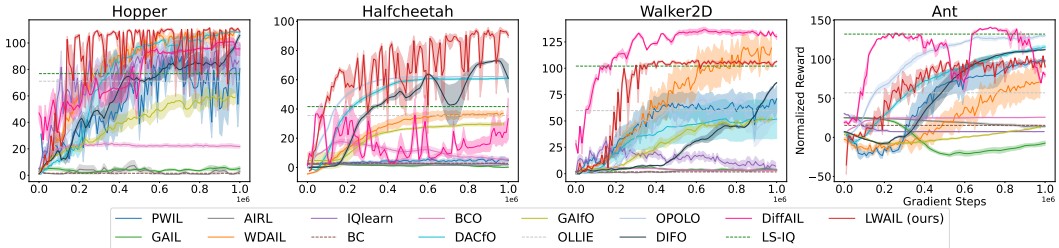

Figure 7: Performance on the MuJoCo environments. Full training curves for all tasks. Note that DiffAIL occasionally collapses on Walker2D; we therefore report the average over successful runs only.

Table 9: Ablation on the size of the random dataset $I$ for ICVF pretraining. Our method works well with a small amount of offline data for ICVF.

| Size | Hopper | HalfCheetah | Walker2D | Ant | Average |
|---|---|---|---|---|---|
| 10K (original) | $108.84 \pm 0.79$ | $90.40 \pm 3.71$ | $106.51 \pm 1.14$ | $90.53 \pm 3.26$ | 99.07 |
| 50K | $92.35 \pm 24.77$ | $89.88 \pm 4.59$ | $104.54 \pm 1.66$ | $89.54 \pm 6.74$ | 94.08 |
| 100K | $109.74 \pm 1.29$ | $91.97 \pm 1.51$ | $100.54 \pm 9.42$ | $73.48 \pm 26.17$ | 93.93 |
| 1M | $110.52 \pm 1.06$ | $86.71 \pm 5.67$ | $105.30 \pm 2.33$ | $80.56 \pm 13.09$ | 95.77 |

## F.2 DATASET ABLATIONS

### F.2.1 ABLATIONS ON THE SIZE OF THE RANDOM DATASET $I$

To investigate the effect of the size of the auxiliary dataset $I$ used for ICVF pretraining, we vary the number of random transitions from 10K up to 1M. Tab. 9 reports the results. We find that our method remains robust even with only 10K random transitions. While performance on some environments fluctuates when the dataset is very large or moderately sized, the overall results indicate that only a small amount of random data is sufficient to achieve strong performance. This confirms the efficiency of our approach in terms of auxiliary data requirements.

### F.2.2 ABLATIONS ON THE QUALITY OF THE RANDOM DATASET $I$

We further examine whether the auxiliary dataset $I$ must be purely random. To this end, we replace the random dataset with the D4RL `medium` and `expert` datasets, and report the results in Tab. 10. We observe that our method performs consistently well regardless of whether $I$ is random, medium, or expert. This demonstrates that our approach is fully compatible with existing offline datasets and does not rely on the randomness assumption. Notably, the performance is comparable across different choices of $I$, further validating the robustness and flexibility of our framework.

Table 10: Ablation on the quality of dataset $I$ for ICVF pretraining. Our method works equally well with random or non-random data sources.

| | Hopper | HalfCheetah | Walker2D | Ant | Average |
|---|---|---|---|---|---|
| Random dataset | $108.84 \pm 0.79$ | $90.40 \pm 3.71$ | $106.51 \pm 1.14$ | $90.53 \pm 3.26$ | 99.07 |
| Medium dataset | $106.24 \pm 2.67$ | $90.12 \pm 2.89$ | $103.85 \pm 1.96$ | $85.04 \pm 3.71$ | 96.31 |
| Expert dataset | $108.11 \pm 1.84$ | $89.82 \pm 3.24$ | $106.42 \pm 1.63$ | $84.57 \pm 2.98$ | 97.23 |

### F.2.3 ABLATIONS ON THE SIZE OF THE EXPERT DATASET $E$

To demonstrate the robustness of our method even if the expert data is scarce, we test our method with 5 expert trajectories and the whole expert dataset (1M transitions). Tab. 11 summarizes the results. We observe consistent compelling performance regardless of the number of expert trajectories.

Table 11: Ablation on using multiple trajectories as expert demonstrations. Our method shows consistent expert-level performance regardless of the number of expert demonstrations.

|  | Hopper | HalfCheetah | Walker2D | Ant | Average |
|---|---|---|---|---|---|
| 1 trajectory | $108.84 \pm 0.79$ | $90.40 \pm 3.71$ | $106.51 \pm 1.14$ | $90.53 \pm 3.26$ | 99.07 |
| 5 trajectories | $107.42 \pm 6.83$ | $91.96 \pm 2.34$ | $107.05 \pm 1.42$ | $88.37 \pm 9.87$ | 98.70 |
| 10 trajectories | $108.11 \pm 5.21$ | $92.15 \pm 2.67$ | $106.64 \pm 1.63$ | $89.42 \pm 8.11$ | 99.08 |
| All expert dataset | $109.21 \pm 3.64$ | $94.07 \pm 2.98$ | $104.59 \pm 2.05$ | $90.72 \pm 9.14$ | 99.65 |

### F.2.4 EXPERT DATASET $E$ WITH SUBSAMPLED TRAJECTORIES

To validate the robustness of our policy, we provide results with subsampled expert trajectories, a widely-adopted scenario in many prior works such as PWIL and IQ-learn. Only a small portion of the complete expert trajectories are present. Our subsample ratio is 10, i.e., we take 1 expert state pair out of adjacent 10 pairs. Tab. 12 summarizes the results, which show that 1) our method with subsampled trajectories outperforms Wasserstein-based baselines such as WDAIL (Zhang et al., 2020) and IQlearn (Garg et al., 2021), and 2) the performance of our method is not affected by incomplete expert trajectories.

Table 12: Ablation on subsampled expert trajectories. The result shows that LWAIL is robust to subsampled expert demonstrations and outperforms other baselines with subsampled expert demonstrations.

|  | Hopper | HalfCheetah | Walker2D | Ant | Average |
|---|---|---|---|---|---|
| LWAIL | $108.84 \pm 0.79$ | $90.40 \pm 3.71$ | $106.51 \pm 1.14$ | $90.53 \pm 3.26$ | **99.07** |
| LWAIL w./ subsample | $109.00 \pm 0.46$ | $86.73 \pm 7.02$ | $106.13 \pm 2.47$ | $83.21 \pm 8.80$ | **96.27** |
| WDAIL w./ subsample | $108.21 \pm 4.90$ | $35.41 \pm 2.07$ | $114.32 \pm 2.07$ | $83.87 \pm 10.92$ | 85.45 |
| IQlearn w./ subsample | $60.26 \pm 14.21$ | $4.12 \pm 1.03$ | $8.31 \pm 1.48$ | $5.32 \pm 3.87$ | 19.50 |

## F.3 ALGORITHM COMPONENT ABLATIONS

### F.3.1 DOWNSTREAM RL ALGORITHM

We used TD3 as our downstream RL algorithm rather than other algorithms such as PPO and DDPG with an entropy regularizer. Our choice is motivated by better efficiency and stability, especially because TD3 is an off-policy algorithm which is more robust to the shift of the reward function and our adversarial training pipeline. We ablate this choice of the downstream RL algorithm and show that TD3 outperforms PPO and DDPG in our framework. Tab. 13 summarizes the results.

Table 13: Ablation on downstream RL algorithms. The result shows that TD3 works much better than PPO and DDPG.

|  | Hopper | HalfCheetah | Walker2D | Ant | Average |
|---|---|---|---|---|---|
| LWAIL+TD3 (original) | $108.84 \pm 0.79$ | $90.40 \pm 3.71$ | $106.51 \pm 1.14$ | $90.53 \pm 3.26$ | **99.07** |
| LWAIL+PPO | $65.21 \pm 4.81$ | $1.02 \pm 0.21$ | $24.13 \pm 2.14$ | $9.12 \pm 0.85$ | 24.87 |
| LWAIL+DDPG | $72.32 \pm 21.76$ | $79.52 \pm 5.21$ | $5.86 \pm 3.13$ | $-41.67 \pm 9.37$ | 29.01 |

### F.3.2 SIGMOID REWARD MAPPING

We adopt the sigmoid function to regulate the output of our neural networks for better stability (similar to WDAIL (Zhang et al., 2020)). However, one cannot naively apply the sigmoid to the reward function for better performance. To show this, we compare to TD3 with a sigmoid function applied to the ground-truth reward. The result is illustrated in Tab. 14. The result shows that a naive sigmoid mapping of the reward does not improve TD3 results.

Table 14: Results of TD3 with and without sigmoid applied on the ground-truth reward. The results show that applying the sigmoid function does not yield better performance.

| Environment | Hopper | HalfCheetah | Walker2D | Ant | Maze2D | Average |
|---|---|---|---|---|---|---|
| TD3 | $105.54 \pm 1.32$ | $76.13 \pm 3.85$ | $89.68 \pm 0.84$ | $89.21 \pm 1.90$ | $120.14 \pm 0.32$ | 96.14 |
| TD3+Sigmoid reward | $84.23 \pm 4.86$ | $30.76 \pm 14.33$ | $42.55 \pm 9.07$ | $34.79 \pm 20.01$ | $119.03 \pm 0.18$ | 62.27 |

## F.4    Detailed Performance Analysis

### F.4.1    Hyperparameter Sensitivity analysis

We present additional ablation studies on hyperparameters in Tab. 15. Please refer to Tab. 8 for detailed definitions and default values. While Adversarial Imitation Learning methods are typically sensitive to hyperparameter configurations, our results demonstrate that LWAIL maintains robust performance across a reasonable range of variations.

Notably, we did not perform an extensive grid search to maximize performance for the main results; instead, we adopted standard settings from prior work to ensure a fair comparison. As shown in the table, specific parameter tuning (e.g., update_interval) can yield results even superior to the reported defaults. Regarding the update interval, we observe that while it has a marginal impact on asymptotic performance, it significantly influences training stability and the smoothness of learning curves.

Table 15: Ablation study on hyperparameters. We compare different Learning Rates (LR), discriminator epochs, and discriminator update intervals.

| | Hopper | HalfCheetah | Walker2D | Ant | Average |
|---|---|---|---|---|---|
| LWAIL (Default) | $110.12 \pm 1.05$ | $87.20 \pm 5.60$ | $104.88 \pm 2.40$ | $81.05 \pm 12.90$ | 95.81 |
| Critic LR=3e-3 | $109.15 \pm 1.47$ | $66.58 \pm 8.31$ | $99.25 \pm 2.44$ | $74.50 \pm 14.92$ | 87.37 |
| Critic LR=3e-4 | $108.96 \pm 1.25$ | $85.91 \pm 5.42$ | $103.87 \pm 2.73$ | $83.84 \pm 13.90$ | 95.65 |
| Actor LR=1e-3 | $110.20 \pm 1.42$ | $85.34 \pm 6.89$ | $104.55 \pm 2.67$ | $76.26 \pm 14.11$ | 94.09 |
| Actor LR=1e-4 | $111.52 \pm 1.31$ | $83.76 \pm 6.75$ | $101.85 \pm 2.59$ | $84.35 \pm 12.66$ | 95.37 |
| Disc. LR=3e-3 | $107.47 \pm 1.38$ | $88.21 \pm 6.12$ | $103.62 \pm 2.70$ | $82.72 \pm 14.03$ | 95.51 |
| Disc. LR=1e-4 | $109.25 \pm 1.28$ | $84.66 \pm 5.54$ | $101.12 \pm 2.51$ | $83.48 \pm 12.91$ | 94.63 |
| Update Epoch=10 | $108.15 \pm 1.17$ | $81.12 \pm 5.44$ | $98.00 \pm 2.87$ | $80.59 \pm 14.33$ | 91.97 |
| Update Epoch=50 | $105.82 \pm 1.33$ | $86.85 \pm 6.22$ | $99.26 \pm 2.46$ | $74.97 \pm 13.77$ | 91.73 |
| Update Interval=1k | $109.12 \pm 1.43$ | $88.54 \pm 6.59$ | $102.46 \pm 2.40$ | $85.95 \pm 13.66$ | **96.52** |
| Update Interval=2k | $106.29 \pm 1.19$ | $89.37 \pm 5.77$ | $103.19 \pm 2.28$ | $81.22 \pm 12.84$ | 95.02 |
| Update Interval=8k | $103.12 \pm 1.51$ | $86.79 \pm 6.44$ | $101.41 \pm 2.65$ | $88.67 \pm 13.92$ | 95.00 |

### F.4.2    More Embedding Visualizations

We only visualize our embedding for HalfCheetah and Walker2d in the main paper due to space limits. In Fig. 8, we visualize results for the embedding learned on the Hopper and Ant environments.

### F.4.3    Pseudo-Reward Metric Curve

To validate the effect of using sigmoid and ICVF embedding for our pseudo-reward generated by $f$, we conduct two experiments:

1) Run a standard setting of LWAIL, and compare pseudo-rewards generated by $f$ with the sigmoid function, and pseudo-rewards without the sigmoid function for the MuJoCo environments. This is illustrated in Fig. 9.

2) Run standard LWAIL and LWAIL without ICVF embedding, and compare pseudo-rewards (with the sigmoid function) for the MuJoCo environments. This is illustrated in Fig. 10.

The value is the total reward accumulated over one evaluation trajectory. The result clearly shows that both ICVF-embedding and the sigmoid function are very important for pseudo-reward stability and positive correlation with ground-truth reward.

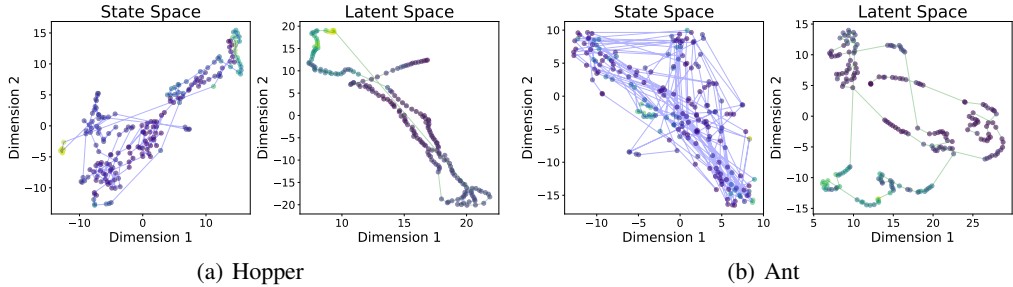

(a) Hopper                     (b) Ant

Figure 8: Visualization of the same trajectory in the original state space and the embedding (latent) space. The color of the points represents the ground-truth reward of the state. We observe that an ICVF-trained embedding provides a much more dynamics-aware metric than the vanilla Euclidean distance. See Fig 2 in the main paper for HalfCheetah and Walker2d environment visualizations.

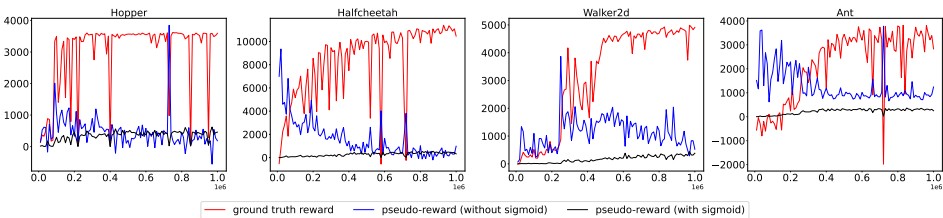

Figure 9: The reward curves of pseudo- and ground-truth reward in a single training session, where pseudo-reward is generated by $f$ following Alg. 1 and serves as the reward signal for our downstream TD3. We note that the pseudo-reward is much more stable and positively correlated with the ground-truth reward when using a sigmoid function.

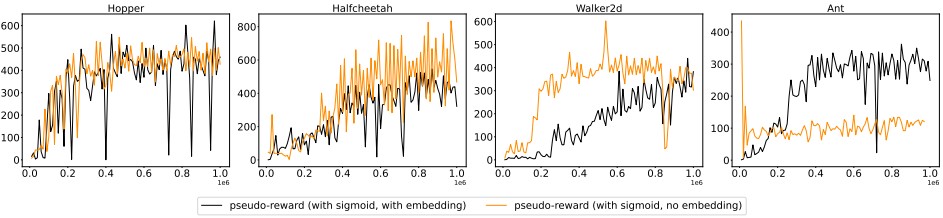

Figure 10: The pseudo-reward curves with and without ICVF embedding in a single training session. We note that without ICVF, the pseudo-reward is generally less stable (e.g., fluctuation in halfcheetah and sudden drop in walker2d and ant) and sometimes less correlated with ground-truth reward (e.g., ant environment).

### F.4.4 COMPARISON TO GROUND-TRUTH REWARD SIGNAL

To show the validity of our learned reward signal, we compare our method with ground-truth guided TD3. Mean and standard deviation are obtained from three independent runs with different seeds. The result is illustrated in Fig. 11. The result shows that LWAIL with our learned reward can achieve comparable or better performance than a human-designed ground-truth reward.

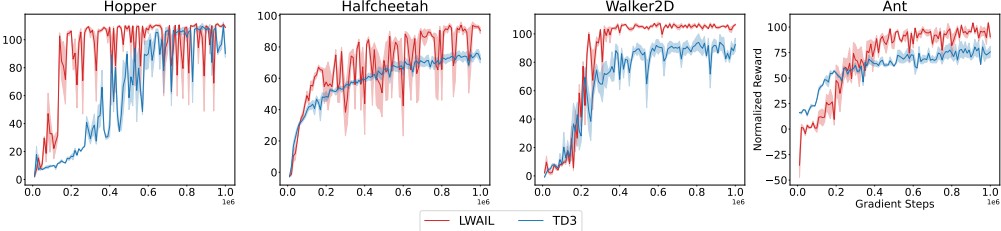

Figure 11: Ablations of the pseudo reward learned by the discriminator and the ground-truth reward. Our learned reward (red) performs equal to or better than TD3 (blue) with ground-truth reward.

## F.5 BASELINE ANALYSIS

### F.5.1 OTHER METHODS WITH LONGER TRAINING

As we use an extra (albeit very small) random offline dataset $I$, raises a potential concern: is the comparison of our method unfair to baselines with purely online imitation learning? To address such concern, we use more steps for several baselines and compare the performance of the baselines and our method. This is illustrated in Tab. 16. We observe that our method still works better than baselines which use more offline (for OLLIE) or online (for online imitation learning methods) data.

| Environment | Hopper | HalfCheetah | Walker2D | Ant | Average |
|---|---|---|---|---|---|
| OLLIE (1M offline + 0.1M online) | $71.10 \pm 3.5$* | $35.50 \pm 4.0$* | $59.80 \pm 8.5$* | $57.10 \pm 7.0$* | 55.87* |
| OPOLO (2M online) | $95.05 \pm 6.50$ | $53.15 \pm 8.90$ | $43.25 \pm 15.10$ | $119.20 \pm 4.65$ | 77.66 |
| DACfO (2M online) | $101.30 \pm 0.58$ | $65.05 \pm 1.05$ | $40.70 \pm 42.00$ | $106.10 \pm 12.50$ | 78.13 |
| DIFO (2M online) | $95.14 \pm 14.30$ | $73.87 \pm 17.86$ | $98.07 \pm 13.88$ | $87.62 \pm 10.75$ | 88.68 |
| DiffAIL (2M online) | $108.56 \pm 0.07$ | $24.38 \pm 21.45$ | $73.06 \pm 19.40$ | $114.62 \pm 4.73$ | 80.16 |
| LWAIL (0.01M offline + 1M online) | $108.84 \pm 0.79$ | $90.40 \pm 3.71$ | $106.51 \pm 1.14$ | $90.53 \pm 3.26$ | 99.07 |
| LWAIL (1M offline + 1M online) | $110.52 \pm 1.06$ | $86.71 \pm 5.67$ | $105.30 \pm 2.33$ | $80.56 \pm 13.09$ | 95.77 |

Table 16: Ablations of LWAIL and baselines trained with more steps (LWAIL with 1M offline uses 1M offline random data, and 0.01M uses 0.01M offline data). Results with * are reported from the main paper. The result clearly shows that our method still works better than baselines trained to see an equal number of or even more data. Note, as DiffAIL consistently collapses before 2M steps, we use the average of the longest runs.

## F.6 MORE ENVIRONMENTS

### F.6.1 BALL-IN-CUP ENVIRONMENT

To test our method on more diverse environments, we report the performance of our method on the ball-in-cup environment of the DeepMind control task (Tassa et al., 2018). This is an 8-dim state and 2-dim action task where the agent needs to put a ball into a connected cup, as illustrated in Fig. 12.

We report the performance of our method with and without ICVF in Tab. 17. The result shows that our method works well on this task, and ICVF plays a vital role.

### F.6.2 MISMATCHED DYNAMICS

It is worth noting that the very motivation of LWAIL is to find a latent space that aligns well with the environment's true dynamics. Despite this, we agree that there might be cases where the latent space employed in LWAIL does not align with the true dynamics due to inaccurate data, e.g., mismatched

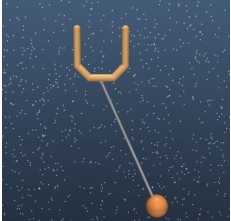

Figure 12: An illustration of the ball-in-cup environment.

| LWAIL | LWAIL w./o. ICVF |
|---|---|
| $84.85 \pm 3.55$ | $14.80 \pm 4.52$ |

Table 17: Normalized reward (0-100 as in the main paper, higher is better) of LWAIL with and without ICVF on the ball-in-cup environment. ICVF plays a vital role on this environment.

dynamics between expert demonstrations and the actual environment. To test such cases, we use the halfcheetah mismatched experts scenario analyzed in SMODICE (Ma et al., 2022): for expert demonstration, the torso of the cheetah agent is halved in length, thus causing inaccurate alignment. We compared our method with the results reported in the SMODICE paper. Tab. 18 summarizes the final average normalized reward (higher is better). Results show that 1) our method works better than several baselines including SMODICE; and 2) our method is robust to mismatched dynamics.

Table 18: Performance on the Halfcheetah environment with mismatched dynamics. Our method outperforms baselines.

|  | Normalized Reward |
|---|---|
| LWAIL | $\mathbf{24.31 \pm 4.51}$ |
| SMODICE | $\mathbf{23.2 \pm 7.43}$ |
| SAIL | $0 \pm 0$ |
| ORIL | $2.47 \pm 0.32$ |

## G    LIST OF NOTATIONS

Tab. 19 summarizes the symbols that appear in our paper.

## H    COMPUTATIONAL RESOURCES

All our experiments are performed with an Ubuntu 20.04 server, which has 128 AMD EPYC 7543 32-Core Processor and a single NVIDIA RTX A6000 GPU. The overall training time of our method is comparable to, or even faster than, that of existing methods. Specifically, ICVF training takes approximately 80–90 minutes, while online training requires about 65–75 minutes for MuJoCo environments. In contrast, PWIL requires 8–9 hours to complete, and DACfO takes 5–6 hours.

Table 19: A list of symbols used in the paper. The first part focuses on RL-specific symbols. The second part details Wasserstein-specific notation. The third part summarizes ICVF-specific symbols (Sec. 3.2).

| Name | Meaning | Note |
|---|---|---|
| $\mathcal{S}$ | State space | |
| $s$ | State | $s \in \mathcal{S}$ |
| $\mathcal{A}$ | Action space | |
| $a$ | Action | $a \in \mathcal{A}$ |
| $t$ | Time step | $t \in \{0, 1, 2, \dots\}$ |
| $\gamma$ | Discount factor | $\gamma \in [0, 1)$ |
| $r$ | Reward function | $r(s, a)$ for single state-action pair |
| $P$ | Transition | $P(s'|s, a) \in \Delta(\mathcal{S})$ |
| $E$ | Expert dataset | state-only expert demonstrations |
| $I$ | Random dataset | state-action trajectories of very low quality |
| $\pi$ | Learner policy | The policy we aim to optimize |
| $d_s^\pi$ | State occupancy of $\pi$ | $d_s^\pi(s) = (1-\gamma)\sum_{i=0}^{\infty} \gamma^i \Pr(s_i = s)$, where $s_i$ is the $i$-th state in a trajectory |
| $d_{ss}^\pi$ | State-pair occupancy of $\pi$ | $d_s^\pi(s, s') = (1-\gamma)\sum_{i=0}^{\infty} \gamma^i \Pr(s_i = s, s_{i+1} = s')$, where $s_i$ is the $i$-th state in a trajectory |
| $d_{ss}^E$ | State-pair occupancy of expert policy | The expert policy here is empirically induced from $E$ |
| $c$ | Underlying metric for Wasserstein distance | |
| $f$ | Dual function / Discriminator | Dual function in Rubinstein dual form of 1-Wasserstein distance; also a discriminator from adversarial perspective and a reward model from IRL perspective |
| $\Pi$ | Wasserstein matching variable | In our case, $\sum_{s \in \mathcal{S}} \Pi(s, s') = d_s^E(s')$, $\sum_{s' \in \mathcal{S}} \Pi(s, s') = d_s^\pi(s)$ |
| $\mathcal{W}_1$ | 1-Wasserstein distance | |
| $s_+$ | Outcome state | |
| $z$ | Latent intention | |
| $V$ | Value function | takes $s, s_+, z$ as input in ICVF; only takes $s$ in normal RL |
| $V_{\text{target}}$ | Target value | target value function in the critic objective of RL |
| $\mathbb{I}$ | indicator function | $\mathbb{I}[\text{condition}] = 1$ if the condition is true, and $= 0$ otherwise |
| $\phi$ | State representation (embedding) | the embedding function we use for $f$; $\phi(s) \in \mathbb{R}^d$ |
| $T$ | Counterfactual intention | $T(z) \in \mathbb{R}^{d \times d}$ |
| $\psi$ | Outcome representation | $\psi(s_+) \in \mathbb{R}^d$ |
| $\alpha$ | ICVF constant | $\alpha \in (0.5, 1]$ |
| $\sigma$ | Sigmoid function | |

