# OpenReview forum: "Latent Wasserstein Adversarial Imitation Learning"
_ICLR.cc/2026/Conference — ICLR 2026 Poster_

### Official Review · Reviewer_Cuqz · 2025-10-14

**Soundness:** 2
**Presentation:** 2
**Contribution:** 2
**Rating:** 4
**Confidence:** 4

**Summary:**

The paper introduces Latent Wasserstein Adversarial Imitation Learning (LWAIL), a imitation learning framework that learns from state-only expert demonstrations without requiring expert actions or large datasets. LWAIL leverages the Wasserstein distance computed in a dynamics-aware latent embedding, which is obtained through a pre-training stage using an Intention Conditioned Value Function (ICVF) trained on a small set of random state-only data. Experiments on multiple MuJoCo tasks show that LWAIL outperforms prior imitation learning methods.

**Strengths:**

- The paper is well-written and easy to follow.
- The experimental results show strong performance compared to both classic imitation learning and learning-from-observation (LfO) methods.
- The ablation studies and experimental details seem to be thorough and sufficient.

**Weaknesses:**

- **Novelty**: The paper primarily combines existing methods, as both WAIL and ICVF are established approaches. This limits the overall technical contribution and originality of the work.

- **Lack of Justifications**: The paper does not sufficiently justify why using the ICVF dynamics-aware embedding space should improve performance. From my understanding, standard adversarial imitation learning (AIL) methods based on occupancy matching are already dynamics-aware—the reward or Q-function trained via the adversarial objective implicitly captures the dynamics, as noted in the original IQ-Learn paper [1]. It remains unclear why introducing an additional dynamics-aware latent embedding would further enhance performance. Moreover, while the t-SNE plot in Figure 2 shows some structure in the learned embedding, its connection to the actual system dynamics is not well established, making it difficult to support the claim that the embedding provides a more dynamics-aware metric.

- **Lack of Comparison with Model-based Approaches**: Several existing model-based imitation learning methods also learn latent embeddings through a latent world model, such as [2, 3, 4]. The paper does not provide sufficient discussion or comparison regarding how the proposed dynamics-aware latent space differs from or improves upon these approaches.

- **Potential Misalignment Issue**: The reward function is trained by discriminating between state-action transitions from expert and random policies. During the RL training phase, however, the policy gradually deviates from the random policy, leading to a potential distribution mismatch. The paper appears to overlook this issue and could benefit from an off-policy correction mechanism, such as importance sampling.

- **Choice of Downstream RL Method**: The ablation study in Table 10 shows that changing the RL algorithm significantly degrades performance. For PPO, the poor results might be due to the aforementioned policy misalignment issue, as PPO is an on-policy method. However, even with off-policy algorithms like DDPG, performance remains unsatisfactory, raising concerns about the robustness and generalizability of the proposed embedding approach.


[1] Garg, D., Chakraborty, S., Cundy, C., Song, J., & Ermon, S. (2021). Iq-learn: Inverse soft-q learning for imitation. Advances in Neural Information Processing Systems, 34, 4028-4039.

[2] Yin, Z. H., Ye, W., Chen, Q., & Gao, Y. (2022). Planning for sample efficient imitation learning. Advances in Neural Information Processing Systems, 35, 2577-2589.

[3] Li, S., Huang, Z., & Su, H. (2025). Reward-free World Models for Online Imitation Learning. In Forty-second International Conference on Machine Learning.

[4] Kolev, V., Rafailov, R., Hatch, K., Wu, J., & Finn, C. (2024, June). Efficient imitation learning with conservative world models. In 6th Annual Learning for Dynamics & Control Conference (pp. 1777-1790). PMLR.

**Questions:**

1. **Justification of Dynamics-aware Embeddings**: My primary concern is the lack of justification for the effectiveness of the proposed dynamics-aware embeddings. Could the authors provide stronger theoretical insights or additional empirical evidence, such as a more detailed analysis of Figure 2 or an ablation study using original observations as inputs to WAIL, to demonstrate that the dynamics-aware embedding is indeed more effective than using raw observations?

2. **Comparison with Latent World Models**: How does the proposed dynamics-aware latent space improve upon the latent representations learned in existing imitation learning methods that employ latent world models?

3. **Sensitivity to Downstream RL Algorithms**: The results show significant performance degradation when changing the downstream RL algorithm. Could the authors provide more justification for this sensitivity? Are there potential robustness issues with the learned embeddings that may contribute to this behavior?

---

> ### Author Response · Authors · 2025-11-24
> **Rebuttal to Reviewer Cuqz (1/3)**
>
> Thanks for your review and constructive feedback. Below are our responses:
>
> **Q1. Novelty issue.** (Weakness 1)
>
> We thank the reviewer for the thoughtful comment. While our method indeed builds on existing components (Wasserstein AIL and ICVF embeddings), our contribution is *not a simple combination of known parts*. Rather, the core contribution lies in *a key insight about their interaction* that addresses a fundamental limitation in prior Wasserstein AIL methods.
>
> 1. **Our contribution goes beyond integrating known methods.**
>
> A central insight of our work is that ICVF embeddings induce a geometry in which Euclidean distance meaningfully reflects dynamical similarity between states. This directly resolves the core challenge in prior Wasserstein AIL using KR duality, where the Euclidean distance in raw state space is misaligned with dynamics. This issue has been discussed in prior works [1,2], but, to the best of our knowledge, no prior AIL method has introduced a principled solution within the dual Wasserstein formulation. PW-DICE [2] and PWIL [3] avoid this weakness by switching to primal formulations and adding additional surrogates, but do not solve the geometry-alignment issue. Thus, *our contribution is the first to introduce a remedy for this known limitation, leading to a cleaner, more principled Wasserstein AIL objective*.
>
> 2. **Our setting and objective are novel and technically meaningful.**
>
> Beyond the insight above, our work also departs from prior AIL literature in important ways:
>
> - Our embedding targets a highly challenging and underexplored regime: *only 10K transitions of low-quality, observation-only data*, potentially from different embodiments and action spaces, whereas prior work typically assumes 100K-1M *state-action* pairs.
>
> - Our formulation minimizes *state-pair transition distributions* within Wasserstein AIL, which, to our knowledge, has not been done before.
>
> - As noted by Reviewer mxsi, our method achieves *expert-level performance using a single state-only expert trajectory* in locomotion tasks, underscoring both the difficulty and practical importance of our setting.
>
> These aspects reflect substantive novelty in both problem formulation and technical perspective, even though the components are individually known.
>
> **Q2. Justifications for dynamic-aware embedding.** (Weakness 2, Question 1)
>
> We answer this question in two parts:
>
> **Part 1. Why dynamic-aware embedding can further improve performance beyond Q-function?**
>
> 1. While with infinite data and perfect convergence, transition-distribution matching does encode optimal paths, since the Q-function (as the dual function for occupancy distributions) “implicitly captures the dynamics” by propagating TD-error through state transitions.  However, this idealization **does not remove the practical need for a geometry-aware embedding** in real policy learning. Two reasons are central:
>
> (a) **Neural networks struggle when semantically different states are numerically close:** If two dynamically distinct states are nearly identical in raw Euclidean space, the value function cannot reliably separate them. A good embedding makes these differences linearly separable and substantially eases optimization.
>
> (b) **Wasserstein AIL fundamentally depends on the quality of the underlying metric:** Unlike f-divergences, Wasserstein distances are defined directly through the metric structure. A poor metric yields a discontinuous, highly non-smooth loss landscape. For example, suppose we use a distance metric where the distance between state pairs is only $0$ if the two state pairs are exactly the same, and $1$ otherwise; the optimal solution remains the same, but the loss landscape induced by such distance metric is apparently much harder to navigate. Enforcing Lipschitzness with non-Euclidean metrics is difficult in practice; embedding-based Euclidean metrics provide a practical and effective solution.
>
> Thus, even though Q-function somewhat encodes dynamics when considering the optimal solution, **in practice, a good state embedding is crucial for learnability, stability, and sample efficiency** under Wasserstein AIL.
>
> 2. Our motivation is directly supported by multiple closely related works (AILOT [4] and PW-DICE), as better state embedding is one of the core motivations in those works which has been proved successful.
>
> 3. The empirical result requested by the reviewer in question 1, “using original observations as inputs to WAIL” is already presented in Tab. 2 of Sec. 4.3 in our original submission. Note, not only our embedding improves the performance, other previous embedding methods also help performance.
>
> (Continued in next part)

---

> ### Author Response · Authors · 2025-11-24
> **Rebuttal to Reviewer Cuqz (2/3)**
>
> (Continued from last part)
>
> **Part 2: Why is ICVF embedding a good choice for dynamic-awareness beyond t-SNE plots in Fig. 2?**
>
> 1. Theoretically, ICVF is optimized based on offline RL that maximizes conditioned future state visit probability, which well-aligns with our state occupancy minimization objective. As shown in **Thm. 3.1**, ICVF captures multi-step transition structure rather than only local one-step relations, providing an embedding that is inherently dynamics-aware. In contrast, , PW-DICE uses contrastive learning with only *adjacent* state as positive anchors, which ignores multi-step transition and trajectory stitching within the dataset; CURL [5] uses contrastive learning to keep embedding consistent against different data augmentation, which could potentially also discriminate numerically similar but dynamically different states; however, it is also not transition- or trajectory-aware.
>
> 2. Regarding the data regime, almost all imitation from observation methods require non-expert state-action pairs (e.g., LobsDICE [6] and OPOLO [7]). ICVF, however, requires only non-expert state transitions with no action needed, which is crucial for our observation-only setting. This makes ICVF uniquely suitable among available embedding approaches.
>
> 3. Regarding the reviewer’s concern about the connection between the t-SNE visualization and the underlying system dynamics: we agree that t-SNE is primarily a qualitative tool and does not by itself prove dynamical correctness. Our intent was to provide intuition that the learned ICVF embedding organizes states in a manner that reflects meaningful behavioral structure. Importantly, the connection to system dynamics is established quantitatively through the downstream control performance, not through t-SNE alone. The embedding is learned by predicting future value under the environment’s transition operator, which explicitly ties it to the dynamics; the strong policy performance across tasks further confirms that the embedding captures the relevant dynamical geometry. We have clarified this distinction in the revision, and verified this in Tab. 4 of Sec. 4.4, where we also compare our embedding with CURL and PW-DICE embedding, which shows that our embedding works better.
>
> **Q3. Comparison with model-based / latent-world model approaches.** (Weakness 3, Question 2)
>
> 1. Thanks for pointing these papers out. Many model-based or latent-world model approaches, such as EffectiveImitate and IQ-MPC (S. Li et al.) mentioned by the reviewer, are not applicable in our scenario for three reasons: 1) they require non-expert state-action pair, while our ICVF only requires non-expert state pairs with no action needed; 2) they require expert action, while our goal is to learn from expert observation only; and 3) they rely on planning-based rollout, which assumes the ability to back up to earlier steps for environments. As for the other paper mentioned by the reviewer, CMIL (V. Kolev et al.), it also requires non-expert action and expert action, and further uses the DreamerV2 architecture for world models, which is hard to adapt to our MuJoCo environment with proprioceptive states. We have added discussion to the revised version of our paper in the “extended related work” section in Appendix.
>
> 2. We would like to further point out that one of our baselines, BCO [8], is a model-based method which first learns the dynamic model through non-expert state-action data, and then conducts imitation learning with the pseudo-labeled expert action. Our model far outperforms BCO as shown in the main results.
>
> **Q4. Potential misalignment issue.** (Weakness 4)
>
> Thank you for raising this point.
>
> 1. The reviewer states that “the reward function is trained by discriminating between **state-action transitions** from **expert and random** policies.” We would like to respectfully clarify the two misunderstandings here. First, we do not utilize or recover any expert action in our algorithm, and are instead discriminating between **state-pair transitions** as shown in Eq. 5. Second, the reward is not discriminating between the expert and random policy; instead, it is discriminating between the expert and the learner’s policy as shown in Eq. 6.
>
> 2. The impact of distribution mismatch on our method is limited because the downstream RL backbone is **TD3**, which is an *off-policy* algorithm that learns from a replay buffer with transitions not sampled from the current policy. Empirically, we observe that the learned ICVF-based reward generalizes sufficiently well across the policy-induced states.
>
> Overall, while the mismatch concern is conceptually relevant, its practical effect is minor for TD3, and incorporating importance sampling would introduce additional variance without clear benefit.

---

> ### Author Response · Authors · 2025-11-24
> **Rebuttal to Reviewer Cuqz (3/3)**
>
> **Q5. Choice of Downstream RL Method.** (Weakness 5, Question 3)
>
> Thank you for the comment. The ablation in our current Table 13 is meant to show that downstream RL algorithms have very different levels of stability when optimizing learned rewards. PPO performs poorly as expected due to on-policy distribution mismatch, and DDPG is known to be significantly less stable and less sample-efficient than TD3 in such settings due to over-exploitation of inaccurate Q-value peaks, Q-value overestimation, and unstable learning dynamics due to critic updated too often (which are the motivation for the core improvements of TD3 over DDPG; see the tutorial [9] for details). Therefore, the performance drop reflects the limitations of these RL backbones rather than a failure of our embedding. When paired with a stable off-policy method like TD3, the proposed latent space consistently yields strong results.
>
> **References**
>
> [1] J. Stanczuk et al. Wasserstein gans work because they fail (to351 approximate the wasserstein distance). arXiv:2103.01678.
>
> [2] K. Yan et al. Offline Imitation from Observation via Primal Wasserstein State Occupancy Matching. In ICML, 2024.
>
> [3] R. Dadashi et al. Primal wasserstein imitation learning. In ICLR, 2021.
>
> [4] M. Borbin et al. Align Your Intents: Offline Imitation Learning via Optimal Transport. In ICLR, 2025.
>
> [5] A. Srinivias et al. CURL: Contrastive Unsupervised Representations for Reinforcement Learning. In ICML, 2020.
>
> [6] G.-H. Kim et al. LobsDICE: Offline Learning from Observation via Stationary Distribution Correction Estimation. In NeurIPS, 2022.
>
> [7] Z. Zhu et al. Off-Policy Imitation Learning from Observations. In NeurIPS, 2020.
>
> [8]  F. Torabi et al. Behavioral Cloning from Observation. In IJCAI, 2018.
>
> [9] OpenAI. Twin Delayed DDPG - Spinning Up documentation. 2018.

---

> > ### Comment · Reviewer_Cuqz · 2025-11-24
> >
> > Thank you for the clarifications. Many of my concerns have been addressed, and I apologize for any misunderstandings in my previous reviews. I do, however, have a few remaining questions:
> >
> > 1. The proposed method with dynamics-aware embeddings is reminiscent of prior work on bisimulation metrics (e.g., [1]). Given the similarity in their objectives, could the authors elaborate on the connections between these approaches, perhaps with additional theoretical justification?
> >
> > 2. While I am not aware of prior work that minimizes the state-pair transition distribution using a Wasserstein formulation in AIL, some existing works (e.g., [2]) minimize this distance using a DICE-based framework. Could the authors provide additional discussion on the relationship to these methods?
> >
> > Thank you again for the clarifications. I will adjust my scores accordingly once these questions are addressed.
> >
> >
> > [1] Castro, P. S. (2020, April). Scalable methods for computing state similarity in deterministic markov decision processes. In Proceedings of the AAAI Conference on Artificial Intelligence (Vol. 34, No. 06, pp. 10069-10076).
> >
> > [2] Sikchi, H., Chuck, C., Zhang, A., & Niekum, S. (2024). A dual approach to imitation learning from observations with offline datasets. arXiv preprint arXiv:2406.08805.

---

> > > ### Author Response · Authors · 2025-11-26
> > > **Response to Reviewer Cuqz's Follow-up (1/2)**
> > >
> > > We appreciate the reviewer’s timely reply. Here are our responses:
> > >
> > > **Q1. The connection of ICVF embedding and bisimulation metrics.**
> > >
> > > We appreciate the reviewer’s insight; the ICVF embedding is indeed connected with the bisimulation metrics to some extent.
> > > **Bisimulation metric** trains an encoder such that in its embedding space, the distance between the embedding approximately equals the bisimulation metric, which is “reward difference + Wasserstein distance of transition distribution in the embedding space.” In the paper provided by the reviewer, the “distance between the embedding” is approximated by another neural network $\psi$, and Wasserstein distance is bypassed by the deterministic property of MDP; in [1], the “distance between the embedding” is Manhattan distance, and Wasserstein distance is $2$-Wasserstein for a convenient closed form.
> > >
> > > **ICVF embedding** trains a value function $V(s,s_+,z)$ conditioned on intention (goal state) $z$ and reward function being the visit probability of the state $s_+$. $V(s,s_+,z)$  is designed to be a product of the embedding and other components.
> > > They are similar as they:
> > >
> > > (1) both use Wasserstein distance (ICVF itself does not use Wasserstein distance, but LWAIL does);
> > >
> > > (2) both address the issue described in Fig. 1(a), which describes the similarity of two states regardless of their numerical values. Bisimulation does so by considering embedded state similarity within one step reach (propagated to trajectories during learning), while ICVF does so by discrimination in different values for the value function with the same intention $z$ and the same reward based on $s_+$.
> > >
> > > (3) both produce representations that can generalize to unseen reward functions. ICVF does so by carefully designing the value function architecture, such that all downstream value functions with state-based rewards are linear representations of the embedding; bisimulation does so by encoding all the causal ancestors of the reward under certain conditions (see Thm. 4 in [1] for details).
> > >
> > > The key difference is that: **bisimulation is action-focused, while ICVF is state-focused**. This difference leads to different advantages of the two embeddings: ICVF can work with only the state available, but the notion of “intention” assumes goal-based downstream rewards for generalization guarantee in (3); bisimulation metric requires actions for training, and is designed for denser reward functions (since the metric directly compares reward). The ability of ICVF utilizing low-quality state-only data which is otherwise unusable aligns with our motivation.

---

> ### Author Response · Authors · 2025-11-26
> **Response to Reviewer Cuqz's Follow-up (2/2)**
>
> **Q2. Additional discussion of connection to DICE.**
>
> Distribution Correct Estimation (DICE) framework is a widely studied imitation learning framework that minimizes the state (SMODICE [2]), state-action (OptiDICE [3]), state-pair (LobsDICE [4]) or (s,a,s’)-transition (DILO) [5] distribution divergence between the expert and the learner’s policy. Some DICE works, such as LobsDICE or PW-DICE [6], add regularizing terms with respect to non-expert data for pessimistic principle on out-of-distribution data during offline learning or avoid zero distribution density as a denominator.
>
> The primal objective of DICE is usually a constrained optimization, with divergence minimization as the objective and Bellman equation as the constraint; to convert it into an unconstrained optimization, DICE methods usually resort to dual functions (which is essentially “value function”) using such as Lagrange dual and Donsker-Varadhan representation, solve the value function via bilevel optimization (in practice, the optimization process is often single-level, as the optimal solution for one layer is designed to have closed form), and eventually recovers the policy by weighted behavior cloning with weights based on the “value function.” Another reason for solving the dual form is that state distribution itself is hard to fit for continuous environments.
>
> Given this background, **the key difference between DICE methods and our Wasserstein AIL framework is that DICE methods are not compatible with Wasserstein distance**; DICE methods usually use $f$-divergences, especially KL and $\chi^2$-divergences. This is because the Bellman constraint is a linear one; to turn the optimization into an unconstrained one, the primal objective must be non-linear. Unfortunately, Wasserstein distance is a linear objective, hence incompatible with DICE. For example, the paper that the reviewer mentioned, DILO, uses $\chi^2$-divergence. There are some exceptions, such as PW-DICE and SoftDICE [7]; however, both methods must add additional non-linear regularizer to bypass this issue (for PW-DICE, it is the state-action KL divergence between learner and non-expert data; for SoftDICE, it is an entropy regularizer). In contrast, our current Wasserstein AIL framework does not confront this issue.
>
> We will add these discussions in the revision, and we hope our response can address the reviewer’s remaining concern. We thank the reviewer again for the time devoted to reviewing our paper, and we are happy to discuss more if the reviewer has further questions.
>
> **Reference**
>
> [1] A. Zhang et al. Learning Invariant Representations for Reinforcement Learning without Reconstruction. In ICLR, 2021.
>
> [2] Y. J. Ma et al. Smodice: Versatile offline imitation learning via state occupancy matching. In ICML, 2022.
>
> [3] J. Lee et al. OptiDICE: Offline Policy Optimization via Stationary Distribution Correction Estimation. In ICML,. 2021.
>
> [4] G.-H. Kim et al. LobsDICE: Offline Learning from Observation via Stationary Distribution Correction Estimation. In NeurIPS, 2022.
>
> [5] Sikchi, H. et al.. (2024). A dual approach to imitation learning from observations with offline datasets. In CORL, 2024.
>
> [6] K. Yan et al. Offline Imitation from Observation via Primal Wasserstein State Occupancy Matching. In ICML, 2024.
>
> [7] M. Sun et al. SoftDICE for Imitation Learning: Rethinking Off-policy Distribution Matching. arXiv, 2021.

---

> > ### Comment · Reviewer_Cuqz · 2025-11-26
> >
> > Thank you for your clarification. My concerns have been addressed and I'm happy to raise my score.
> >
> > Best,
> > Reviewer

---

> > > ### Author Response · Authors · 2025-11-26
> > >
> > > We thank the reviewer for the constructive feedback throughout the review process and for the decision to raise the score. We are glad that our responses have addressed all your concerns. We will ensure that these discussions are incorporated into the final revision of the paper.

---

### Official Review · Reviewer_mxsi · 2025-10-26

**Soundness:** 3
**Presentation:** 3
**Contribution:** 3
**Rating:** 6
**Confidence:** 4

**Summary:**

Previous Wasserstein imitation learning methods couldn’t capture the environment’s dynamics accurately. The paper proposes a novel adversarial imitation learning framework, Latent Wasserstein Adversarial Imitation Learning (LWAIL), that integrates the Intention Conditioned Value Function (ICVF) to capture these dynamics to resolve the issue. By using a two-stage process, first pretraining the ICVF with low-quality data and, secondly, running a standard Wasserstein AIL framework in this new latent space, the paper shows expert-level results in multiple locomotion tasks with a single state-only expert trajectory.

**Strengths:**

- The paper is clearly written, well-structured, and comprehensive, making the authors' contributions and methodology easy to follow.
- The paper's primary contribution, LWAIL, is novel in its integration of an ICVF-learned latent space into a Wasserstein AIL framework to create a dynamics-aware distance metric. This is significant as it demonstrates expert-level performance on several locomotion tasks using only a single state-only expert trajectory, a challenging and practical problem setting.
- The empirical investigation is of high quality. The authors performed a thorough comparison against numerous baselines and provided an extensive set of ablation studies that validate the key components of their approach, such as the choice of embedding and robustness to noise.

**Weaknesses:**

- The empirical evaluation, while thorough on the chosen tasks, is limited to locomotion (MuJoCo) and a simple maze environment. These environments are state-based and have relatively stable dynamics. The applicability of LWAIL to more complex, high-dimensional problems (e.g., vision-based tasks) or environments with more stochastic dynamics is not explored.
- The paper does not include an ablation study on the sensitivity of key hyperparameters. Adversarial Imitation Learning frameworks are often sensitive to settings like the discriminator update frequency (which the authors note must be balanced ) or the gradient penalty coefficient. An analysis of how performance changes with these hyperparameters would strengthen the paper's claims of robustness.
- The paper's core motivation, powerfully illustrated in Figure 1(a), is that LWAIL's dynamics-aware metric can solve problems where simple Euclidean distance is misleading (e.g., a state near a goal is "worse" than a farther state with a valid path). However, the chosen experiments, while standard, do not fully stress-test this specific claim. The `umaze` environment demonstrates this principle, but is very simple. The MuJoCo locomotion tasks are complex in terms of control, but they do not necessarily exhibit the challenging topological structure implied by the motivating example. The paper would be much stronger if LWAIL and key baselines were evaluated on environments specifically designed to have this deceptive property, such as more complex mazes with traps, one-way paths, or regions that require non-Euclidean navigation to succeed.

**Questions:**

- The paper convincingly argues that an ICVF-learned latent space is beneficial for tasks like `umaze`, where the Euclidean distance is misleading due to the environment's dynamics (e.g., a wall). However, it is less clear how this specific benefit translates to the MuJoCo locomotion tasks. What is the intuition for why a dynamics-aware latent space is superior to the standard state space for tasks like Hopper or Walker2D?
- To better validate the core contribution, have you considered testing LWAIL and key baselines on more complex environments that explicitly share the properties of Figure 1(a), such as the `ant-maze` or other complex navigation tasks from the D4RL benchmark?
- The performance plot in Figure 4 is very dense, and many of the colors are similar, making it difficult to distinguish the learning curve for LWAIL from the baselines. Could you please consider using a more distinct color (e.g., a thick, bright red) or a different line style for your method in the final version to improve legibility?

---

> ### Author Response · Authors · 2025-11-24
> **Rebuttal to Reviewer mxsi (1/2)**
>
> Thanks for your review and constructive feedback. Below are our responses:
>
> **Q1. More complex, high-dimensional problems.** (Weakness 1)
>
> To address the reviewer’s concern, we have included a series of additional experiments in the appendix on more challenging environments. All scores are normalized to 0-100 by dividing the reward with the expert trajectory’s reward (Maze2d are normalized by D4RL settings); the higher is better.
>
> **Humanoid-v5**
>
>  Method        | Score
> ---|---
> LWAIL         | 24.36 ($\pm$ 2.53)
> LWAIL without  ICVF  | 7.13 ($\pm$ 0.67)
>
> **AntMaze-Umaze-v2**
>
> Method        | Score
> ---|---
> LWAIL         | 34.76 ($\pm$ 7.21)
> LWAIL without ICVF | 0.03 ($\pm$ 0.01)
>
> **DMControl: Ball-in-Cup**
>
>  Method        | Score
> --|---
>  LWAIL         | 84.85 ($\pm$ 3.55)
> LWAIL without ICVF | 14.80 ($\pm$ 4.52)
>
> We also conduct experiments on the maze2d-medium and maze2d-large, and add different stochasticity (noise level) on the initial state. Below is the result:
>
> **maze2d-medium**
>
> Noise Level | LWAIL            | Without LWAIL
> ---|---|---
>  0.0         | 144.59 ($\pm$ 0.87)          | 117.91 ($\pm$ 0.30)
>  0.2         | 143.42 ($\pm$ 0.91)          | -17.00 ($\pm$ 0.01)
>  0.5         | 135.12 ($\pm$ 3.12)          | -16.98 ($\pm$ 0.02)
>
> **maze2d-large**
>
> Noise Level | LWAIL            | Without LWAIL
> ---|---|---
>  0.0         | 156.68 ($\pm$ 0.60)          | 148.25($\pm$ 6.41)
>  0.2         | 157.04 ($\pm$ 0.71)          | -11.11 ($\pm$ 0.04)
>  0.5         | 157.06 ($\pm$ 0.81)          | -11.13 ($\pm$ 0.02)
>
> These results demonstrate that our method remains robust and effective in higher-dimensional, long-horizon, and more complex control settings, supporting the generality of our approach beyond low-dimensional locomotion.
>
> **Q2. The sensitivity of key hyperparameters.** (Weakness 2)
>
> We provide additional hyperparameter sensitivity experiments as shown below. Please refer to Table 8 in the paper for the definition of each hyperparameter and their default values used in our main experiments. While it is known that AIL methods can be sensitive to certain hyperparameters, we find that our method performs consistently well across a wide and reasonable range of these values. The variations produce only modest changes in performance, indicating that our approach is comparatively robust despite operating in a challenging observation-only setting.
>
> We also emphasize that we did not perform an extensive grid search to tune hyperparameters for the best possible performance (better results are shown in the table below). Instead, we largely followed the settings used in prior work, aiming to demonstrate the improvements brought by our method under comparable and fair experimental conditions. For the update interval parameter, although it does not affect the final performance, it can introduce noticeable spikes in the learning curves.
>
>
> Env                | Hopper             | HalfCheetah       | Walker             | Ant                | Average
> --- | --- | --- | --- | --- | ---
>  Default             | 110.12 ($\pm$1.05) | 87.20 ($\pm$5.60) | 104.88 ($\pm$2.40) | 81.05 ($\pm$12.90) | 95.81
>  Critic LR=3e-3     | 109.15 ($\pm$1.47) | 66.58 ($\pm$8.31) | 99.25 ($\pm$2.44)  | 74.50 ($\pm$14.92) | 87.37
>  Critic LR=3e-4     | 108.96 ($\pm$1.25) | 85.91 ($\pm$5.42) | 103.87 ($\pm$2.73) | 83.84 ($\pm$13.90) | 95.65
>  Actor LR=1e-3      | 110.20 ($\pm$1.42) | 85.34 ($\pm$6.89) | 104.55 ($\pm$2.67) | 76.26 ($\pm$14.11) | 94.09
>  Actor LR=1e-4      | 111.52 ($\pm$1.31) | 83.76 ($\pm$6.75) | 101.85 ($\pm$2.59) | 84.35 ($\pm$12.66) | 95.37
>  Disc. LR=3e-3          | 107.47 ($\pm$1.38) | 88.21 ($\pm$6.12) | 103.62 ($\pm$2.70) | 82.72 ($\pm$14.03) | 95.51
>  Disc. LR=1e-4          | 109.25 ($\pm$1.28) | 84.66 ($\pm$5.54) | 101.12 ($\pm$2.51) | 83.48 ($\pm$12.91) | 94.63
> Disc. Update Epoch=10         | 108.15 ($\pm$1.17) | 81.12 ($\pm$5.44) | 98.00 ($\pm$2.87) | 80.59 ($\pm$14.33) | 91.97
> Disc. Update Epoch=50         | 105.82 ($\pm$1.33) | 86.85 ($\pm$6.22) | 99.26 ($\pm$2.46)  | 74.97 ($\pm$13.77) | 91.73
> Disc. Update Interval=1k | 109.12 ($\pm$1.43) | 88.54 ($\pm$6.59) | 102.46 ($\pm$2.40) | 85.95 ($\pm$13.66) | 96.52
> Disc. Update Interval=2k | 106.29 ($\pm$1.19) | 89.37 ($\pm$5.77) | 103.19 ($\pm$2.28) | 81.22 ($\pm$12.84) | 95.02
> Disc. Update Interval=8k | 103.12 ($\pm$1.51) | 86.79 ($\pm$6.44) | 101.41 ($\pm$2.65) | 88.67 ($\pm$13.92) | 95.00

---

> ### Author Response · Authors · 2025-11-24
> **Rebuttal to Reviewer mxsi (2/2)**
>
> **Q3. More complex mazes.** (Weakness 3, Question 2)
>
> Thank you for the insightful suggestion. To address this, we have conducted additional experiments on navigation tasks using the maze2d-medium and maze2d-large environments, where a point-mass agent must navigate through a complex maze.
>
> To evaluate robustness against distribution shifts, we introduced varying levels of noise to the agent's initialization point for maze2d-medium and maze2d-large. This forces the agent into unseen states that are not present in the expert dataset. The results are presented below:
>
> **maze2d-medium**
> Noise Level | LWAIL            | Without LWAIL
> ---|---|---
>  0.0         | 144.59 ($\pm$ 0.87)          | 117.91 ($\pm$ 0.30)
>  0.2         | 143.42 ($\pm$ 0.91)          | -17.00 ($\pm$ 0.01)
>  0.5         | 135.12 ($\pm$ 3.12)          | -16.98 ($\pm$ 0.02)
>
> **maze2d-large**
>
> Noise Level | LWAIL            | Without LWAIL
> ---|---|---
>  0.0         | 156.68 ($\pm$ 0.60)          | 148.25($\pm$ 6.41)
>  0.2         | 157.04 ($\pm$ 0.71)          | -11.11 ($\pm$ 0.04)
>  0.5         | 157.06 ($\pm$ 0.81)          | -11.13 ($\pm$ 0.02)
>
> As shown in the tables, LWAIL demonstrates significantly superior robustness compared to the baseline. While the performance of the model without LWAIL degrades catastrophically when noise is introduced (dropping to negative scores), our method maintains high rewards. This indicates that our dynamic-aware embeddings effectively capture underlying dynamics, enabling the agent to generalize well to new observations and unseen states.
>
> We also test LWAIL on antmaze, showing comparable performance to baselines. Note, unlike SMODICE, we do not need to use different divergence (KL vs. chi-square) for different environments; we use the same hyperparameter as MuJoCo environments.
>
> **AntMaze-Umaze-v2**
>
> Method        | Score
> ---|---
> LWAIL         | 34.76 ($\pm$ 7.21)
> LWAIL without ICVF | 0.03 ($\pm$ 0.01)
> WDAIL |  31.03 ($\pm$ 5.62)
> IQ-Learn | 0.00 ($\pm$ 0.00)
> SMODICE | 44.44 ($\pm$ 7.31)
>
> **Q4. Intuitively, why dynamics-aware state embedding benefits locomotion tasks?** (Question 1)
>
> Great question. We argue that the dynamic-aware state embedding gives benefit in the following two aspects:
>
> (a) **Neural networks struggle when semantically different states are numerically close:** If two dynamically distinct states are nearly identical in raw Euclidean space, the value function cannot reliably separate them. A good embedding makes these differences linearly separable and substantially eases optimization.
>
> (b) **Wasserstein AIL fundamentally depends on the quality of the underlying metric:** Unlike f-divergences, Wasserstein distances are defined directly through the metric structure. A poor metric yields a discontinuous, highly non-smooth loss landscape. For example, suppose we use a distance metric where the distance between state pairs is only $0$ if the two state pairs are exactly the same, and $1$ otherwise; the optimal solution remains the same, but the loss landscape induced by such distance metric is apparently much harder to navigate. Enforcing Lipschitzness with non-Euclidean metrics is difficult in practice; embedding-based Euclidean metrics provide a practical and effective solution.
>
> Thus, **in practice, a good state embedding is crucial for learnability, stability, and sample efficiency** under Wasserstein AIL. This intuition is also directly supported by multiple closely related works (AILOT [1] and PW-DICE [2]), as better state embedding is one of the core motivations in those works which has been proved successful. Note, in our comparison to other embedding methods in our current Tab. 4, all embedding methods improve performance on locomotion tasks to some extent.
>
> **Q5. The performance plot in Figure 4 is very dense.** (Question 3)
>
> Thanks for pointing that out; we have already fixed the issue. We keep the important and comparable baseline curves in the main paper, and we have moved the full figure to the Appendix Fig. 7.

---

### Official Review · Reviewer_EztW · 2025-10-29

**Soundness:** 2
**Presentation:** 3
**Contribution:** 2
**Rating:** 2
**Confidence:** 4

**Summary:**

This paper proposes a novel Latent Wasserstein Adversarial Imitation Learning (LWAIL) algorithm for state-only imitation. The authors identify that in Wasserstein IL, the gradient penalty commonly used to stabilize adversarial optimization enforces a 1-Lipschitz constraint, which implicitly restricts the underlying distance metric to be Euclidean. To overcome this limitation, LWAIL introduces a dynamics-aware latent space learned via the Intention-Conditioned Value Function, which captures reachability between states and provides a more meaningful ground metric for Wasserstein distance.

**Strengths:**

The paper identifies and addresses the weakness of using Euclidean metrics in Wasserstein IL by introducing a dynamics-aware latent space. Overall, the presentation is clear, well-structured, and easy to follow.

**Weaknesses:**

1. **Limited novelty**: The contribution is primarily a combination of existing ideas (Wasserstein IL, adversarial learning, and latent representations), with the main advance being the integration of ICVF embeddings.

2. **Narrow experimental scope**: The experiments focus mainly on locomotion tasks. Given the illustrative example in Fig. 1, navigation-style tasks (such as those in Fig. 3) would have been more suitable to highlight the strengths of the proposed approach.

3. **Questionable necessity of state embeddings**: A central motivation of the paper is that Euclidean distances between states are not meaningful. However, in many AIL approaches the divergence is minimized over state transitions rather than individual states. Under the standard expert optimality assumption, these transition distributions already capture the structure of optimal paths, potentially mitigating the issue highlighted in Fig. 1(a). This raises the question of whether a dedicated dynamics-aware embedding is strictly necessary or whether the problem is indirectly addressed by existing formulations.

Minor issues:
1. DACfO in line 339 outperforms LWAIL.
2. The Related Work section is missing relevant references, such as:
Giammarino V, Queeney J, Paschalidis I. Adversarial Imitation Learning from Visual Observations using Latent Information. Transactions on Machine Learning Research.

**Questions:**

1. It is unclear what the expert performance level is in Figure 4. Could you provide a reference line or explicit numbers for comparison?

2. In practice, many AIfO algorithms minimize a divergence over state transitions (as in Eq. (5)) rather than over individual states. Wouldn’t this already mitigate the issue highlighted in Fig. 1(a), since the optimization objective ensures that the agent’s transitions (s,s’) are aligned with the expert’s, regardless of the absolute distance between individual state embeddings? How does the proposed embedding provide an additional benefit beyond what is already achieved by transition-level matching?

---

> ### Author Response · Authors · 2025-11-24
> **Rebuttal to Reviewer Eztw (1/3)**
>
> Thanks for your review and constructive feedback. Below are our responses:
>
> **Q1. Limited novelty.** (Weakness 1)
>
> We thank the reviewer for the thoughtful comment. While our method indeed builds on existing components (Wasserstein AIL and ICVF embeddings), our contribution is *not a simple combination of known parts*. Rather, the core contribution lies in *a key insight about their interaction* that addresses a fundamental limitation in prior Wasserstein AIL methods.
>
> 1. **Our contribution goes beyond integrating known methods.**
>
> A central insight of our work is that ICVF embeddings induce a geometry in which Euclidean distance meaningfully reflects dynamical similarity between states. This directly resolves the core challenge in prior Wasserstein AIL using KR duality, where the Euclidean distance in raw state space is misaligned with dynamics. This issue has been discussed in prior works [1,2], but, to the best of our knowledge, no prior AIL method has introduced a principled solution within the dual Wasserstein formulation. PW-DICE [2] and PWIL [3] avoid this weakness by switching to primal formulations and adding additional surrogates, but do not solve the geometry-alignment issue. Thus, *our contribution is the first to introduce a remedy for this known limitation, leading to a cleaner, more principled Wasserstein AIL objective*.
>
> 2. **Our setting and objective are novel and technically meaningful.**
>
>  Beyond the insight above, our work also departs from prior AIL literature in important ways:
> - Our embedding targets a highly challenging and underexplored regime: *only 10K transitions of low-quality, observation-only data*, potentially from different embodiments and action spaces, whereas prior work typically assumes 100K-1M *state-action* pairs.
>
> - Our formulation minimizes *state-pair transition distributions* within Wasserstein AIL, which, to our knowledge, has not been done before.
>
> - As noted by Reviewer mxsi, our method achieves *expert-level performance using a single state-only expert trajectory* in locomotion tasks, underscoring both the difficulty and practical importance of our setting.
> These aspects reflect substantive novelty in both problem formulation and technical perspective, even though the components are individually known.
>
> **Q2. Narrow experimental scope.** (Weakness 2)
>
> Thank you for the insightful suggestion. To address this, we have conducted additional experiments on navigation tasks using the antmaze, maze2d-medium, and maze2d-large environments, where a point-mass agent must navigate through a complex maze.
>
> To evaluate robustness against distribution shifts, we introduced varying levels of noise to the agent's initialization point for maze2d-medium and maze2d-large. This forces the agent into unseen states that are not present in the expert dataset. The results are presented below:
>
> **AntMaze-Umaze-v2**
>
> Method        | Score
> ---|---
> LWAIL         | 34.76 ($\pm$ 7.21)
> LWAIL without ICVF | 0.03 ($\pm$ 0.01)
>
> **maze2d-medium**
>
> Noise Level | LWAIL            | Without LWAIL
> ---|---|---
>  0.0         | 144.59 ($\pm$ 0.87)          | 117.91 ($\pm$ 0.30)
>  0.2         | 143.42 ($\pm$ 0.91)          | -17.00 ($\pm$ 0.01)
>  0.5         | 135.12 ($\pm$ 3.12)          | -16.98 ($\pm$ 0.02)
>
> **maze2d-large**
>
> Noise Level | LWAIL            | Without LWAIL
> ---|---|---
>  0.0         | 156.68 ($\pm$ 0.60)          | 148.25($\pm$ 6.41)
>  0.2         | 157.04 ($\pm$ 0.71)          | -11.11 ($\pm$ 0.04)
>  0.5         | 157.06 ($\pm$ 0.81)          | -11.13 ($\pm$ 0.02)
>
> As shown in the tables, LWAIL demonstrates significantly superior robustness compared to the results without ICVF. While the performance of the model without LWAIL degrades catastrophically when noise is introduced (dropping to negative scores), our method maintains high rewards. This indicates that our dynamic-aware embeddings effectively capture underlying dynamics, enabling the agent to generalize well to new observations and unseen states.

---

> ### Author Response · Authors · 2025-11-24
> **Rebuttal to Reviewer EztW (2/3)**
>
> **Q3. Questionable necessity of state embeddings.** (Weakness 3, Question 2)
>
> We thank the reviewer for raising this important question. We respectfully clarify that, under the setting of our work, **a dynamics-aware state embedding is indeed necessary**, and existing AIL/AIfO formulations do not implicitly resolve the issue illustrated in Fig. 1(a). Our reasons are as follows.
>
> 1. **ICVF embedding only requires non-expert states.**
>
> The offline data used to train our ICVF consist of **only very low-quality (near-random) non-expert state transitions, not state-action pairs**. This setting is, to our best knowledge, not supported by existing AIfO/ transition-matching IL algorithms that utilize non-expert data. Methods such as OPOLO [5], GAIfO [6], and LS-IQ [7] all require **non-expert state-action data of substantially higher quality** in order to minimize divergences over transition distributions. Because these methods cannot operate under our observation-only, low-quality, action-free data regime, we introduce ICVF specifically to extract meaningful dynamics structure from data that would otherwise be unusable for existing approaches.
>
> 2. **Even when state-action rollouts are available during online policy learning, existing transition-matching approaches still do not eliminate the need for a dynamics-aware embedding.**
>
> We address three relevant categories:
>
> (a) **Many prior methods are still incompatible:** Our work falls under **Learning from Observation (LfO)**, where **only state transitions, not expert actions**, are available. This is a common and practical scenario in robotics (e.g., imitation from videos, cross-embodiment demonstrations). Hence, methods such as V-MAIL [8] and PWIL (LfO version of PWIL does not optimize state transition divergence) are inapplicable with expert actions missing.
>
> (b) **Applicable methods often do not use Wasserstein distance:** Some methods (e.g., LobsDICE [9], OPOLO) match transitions using *f-divergences*, which do not exploit geometric structure and do not benefit from the smoothness properties of the Wasserstein distance (especially when the expert states in $E$ are far from those in the non-expert dataset $I$).
>
> (c) **The few applicable Wasserstein-based approaches actually support our claim:** Among Wasserstein IL works, none minimize state-pair transition distributions as we do. LS-IQ uses an inverse dynamic model to pseudo-label action for Q-values; PWIL, PW-DICE, and OTR [10] optimize only at the state level; OTR and AILOT [11] do not match distributions at all. Notably, both PW-DICE and AILOT explicitly introduce state embeddings to improve distance geometry – directly reinforcing our claim that *distance structure matters and that embedding is necessary for Wasserstein-based IL*.
>
> Thus, no prior method simultaneously satisfies: 1) same dataset setting as ours, 2) Wasserstein geometry, 3) transition-distribution matching, 4) no auxiliary action surrogates. Our method is the first to address this combination, and prior work **supports**, rather than contradicts, the necessity of an embedding.
>
> 3. **Even theoretically, transition-matching alone does not resolve geometric issues in finite-data, nonlinear function approximation regimes.**
>
> While with infinite data and perfect convergence, transition-distribution matching does encode optimal paths, this idealization **does not remove the practical need for a geometry-aware embedding** in real policy learning. Two reasons are central:
>
> (a) **Neural networks struggle when semantically different states are numerically close:** If two dynamically distinct states are nearly identical in raw Euclidean space, the value function cannot reliably separate them. A good embedding makes these differences linearly separable and substantially eases optimization.
>
> (b) **Wasserstein AIL fundamentally depends on the quality of the underlying metric:** Unlike f-divergences, Wasserstein distances are defined directly through the metric structure. A poor metric yields a discontinuous, highly non-smooth loss landscape. Enforcing Lipschitzness with non-Euclidean metrics is difficult in practice; embedding-based Euclidean metrics provide a practical and effective solution.
>
> Thus, even though minimizing transition distributions is theoretically sufficient asymptotically, in practice, a good state embedding is **crucial for learnability, stability, and sample efficiency** under Wasserstein AIL.
>
> 4. **We empirically demonstrates the necessity of state embeddings.**
>
> We corroborate this both qualitatively and quantitatively: 1) *t-SNE analyses (Fig. 2)* show that ICVF produces geometry aligned with dynamics. 2) *Ablations (Tab. 2)* show significant performance drops when removing our embedding or replacing it with CURL, and that PW-DICE’s embedding further supports our claim that embeddings help. These results clearly demonstrate that dynamically aligned geometry substantially improves Wasserstein AIL performance.

---

> ### Author Response · Authors · 2025-11-24
> **Rebuttal to Reviewer EztW (3/3)**
>
> **Question 1 and Minor Issues in Weaknesses.**
>
> 1. DACfO in line 339 outperforms LWAIL: Thank you for the comment. While DACfO is slightly better on a few individual tasks, LWAIL achieves much stronger performance on the majority of environments and with far lower variance. Most importantly, the last row reports the average performance across all tasks: LWAIL scores 99.07, clearly higher than DACfO’s 82.41. This average metric reflects overall capability and shows that LWAIL is more consistent and robust than DACfO.
>
> 2. Missing Reference: Thanks for pointing out; we have added the requested references in our PDF.
>
> 3. Expert performance level is in Figure 4: Thanks for pointing out; we have added the explicit number in the description for better comparison.
>
> **References**
>
> [1] J. Stanczuk et al. Wasserstein GANs Work Because They Fail (to Approximate the Wasserstein Distance). arXiv, 2021.
>
> [2] K. Yan et al. Offline Imitation from Observation via Primal Wasserstein State Occupancy Matching. In ICML, 2024.
>
> [3] R. Dadashi et al. Primal Wasserstein Imitation Learning. In ICLR, 2021.
>
> [4] Y. J. Ma et al. Smodice: Versatile offline imitation learning via state occupancy matching. In ICML, 2022.
>
> [5] Z. Zhu et al. Off-Policy Imitation Learning from Observations. In NeurIPS, 2020.
>
> [6] F. Torabi et al. Generative Adversarial Imitation from Observation. In ICML, 2019.
>
> [7] F. Al-Hafez et al. LS-IQ: Implicit Reward Regularization for Inverse Reinforcement Learning. In ICLR, 2023.
>
> [8] R. Rafailov et al. Visual Adversarial Imitation Learning using Variational Models. In NeurIPS, 2021.
>
> [9] G.-H. Kim et al. LobsDICE: Offline Learning from Observation via Stationary Distribution Correction Estimation. In NeurIPS, 2022.
>
> [10] Y. Luo et al. Optimal Transport for Offline Imitation Learning. In ICLR, 2023.
>
> [11] M. Borbin et al. Align Your Intents: Offline Imitation Learning via Optimal Transport. In ICLR, 2025.

---

> ### Comment · Reviewer_EztW · 2025-11-24
>
> Thank you for the response. I have a few follow-up points:
>
> ### Q1
>
> > Our contribution goes beyond integrating known methods.
>
> I understand the authors’ position, and I agree that showing that ICVF embeddings induce a geometry in which Euclidean distance meaningfully reflects dynamical similarity between states is an interesting result.
>
> I would encourage the authors to highlight this aspect more explicitly in the paper. In particular, Contribution (1) currently reads “ICVF offers a compelling metric,” which in my view does not clearly convey the significance of the result.
>
> -----------------------------------------------------------------------------------------------------------------------
> > **Our setting and objectives are novel and technically meaningful.**
>
> I remain unconvinced by some aspects of the response:
>
> > Our embedding targets a highly challenging and underexplored regime: only 10K transitions of low-quality, observation-only data, potentially from different embodiments and action spaces.
>
> Could the authors clarify what is meant here? As far as I can tell, the embedding is not actually trained on data from different embodiments and action spaces. In addition, the claim that 10K transitions are “low data” is application dependent and might not be universally perceived as such.
>
> > Our formulation minimizes state-pair transition distributions within Wasserstein AIL.
>
> While I appreciate the connection to the Wasserstein AIL framework, I would note that state-only AIL itself is a well-established line of work.
>
> > Our method achieves expert-level performance using a single state-only expert trajectory in locomotion tasks.
>
> Other methods have also demonstrated expert-level performance from a single state-only or pixel-based expert trajectory; please see the reference mentioned in my previous comment (Giammarino et al. TMLR, Fig. 9).
>
> ### Q2
>
> I appreciate the additional experiments, which are now more aligned with the main narrative of the paper.
>
> ### Q3
>
> > ICVF embedding only requires non-expert states.
>
> This answer is still confusing to me. While ICVF may indeed require only non-expert states for learning the embedding, the subsequent imitation learning stage still requires at least one state-only expert trajectory, and this trajectory must be of reasonably high quality. I therefore do not fully understand why the authors emphasize the “non-expert only” requirement for the embedding and then refer to GAIfO: GAIfO needs high-quality data precisely because it is an imitation learning algorithm, not a method for learning embeddings.
>
> ### Other Points
>
> I understand that, within the Wasserstein AIL setting, the proposed embeddings are a natural fit. However, in other AIL formulations, such as those based on a BCE loss (e.g., GAIfO or DACfO), such an embedding is not strictly necessary to enable learning, correct?
>
> This raises a follow-up question: why should one prefer the Wasserstein distance in this context? This seems somewhat under-discussed in the paper. The only explicit mention I noticed is in Lines 42-43: “To match distributions within LfO techniques, the Wasserstein distance has gained popularity (Arjovsky et al., 2017; Zhang et al., 2020b).”
>
> I would encourage the authors to expand on this point and more clearly justify the choice of Wasserstein distance over alternative divergences.

---

> > ### Author Response · Authors · 2025-11-26
> > **Response to Reviewer EztW's Follow-up (1/2)**
> >
> > We appreciate the reviewer’s timely reply. We are encouraged that the reviewer finds our core insight regarding the geometry of ICVF embeddings interesting and that the additional experiments on navigation tasks have addressed your concerns. Regarding the remaining concerns, here are our responses:
> >
> > **Q1. Further Clarification of Contributions and Claims**
> >
> > 1. We thank the reviewer for the advice. We now revise Contribution (1) in the PDF to explicitly state: *We show that ICVF embeddings induce a geometry in which Euclidean distance meaningfully reflects dynamical similarity between states, overcoming the geometric limitations of prior Wasserstein occupancy matching methods.*
> >
> > 2. On the "different embodiments and action spaces" claim: our intention was to highlight an additional capability of our method and its potential applications, for which we conducted the mismatched dynamics experiments in Appendix E.6.2. The results demonstrate our method's robustness when the expert’s dynamics differ from the agent’s. We will clarify this point more explicitly in the revision.
> >
> > 3. On the “low data” claim: The use of “low” here has two meanings: “low quality” and “low amount.” For the former, data collected by random policy is generally considered low quality in many offline IL/RL papers, such as OLLIE [1], as opposed to “medium” and “medium-replay” data in the D4RL dataset. For the latter, we agree that in some contexts, 10K transitions are not a small amount. However, we use the word "low" in comparison to many of our baselines referenced in the rebuttal, which typically rely on 100K~1M offline transitions (vs. our 10k). We will add this explicit comparison.
> >
> > 4. On state-only AIL: We agree that state-only AIL is a widely studied area with lots of attention in the community. Our intention was to highlight that our formulation offers a complementary insight specific to the Wasserstein AIL framework. In particular, minimizing state-pair transition distributions allows us to address the metric misalignment in the Wasserstein dual, which is a challenge unique to this setting. We believe this perspective contributes meaningfully to the ongoing development of Wasserstein-based AIL methods.
> >
> > 5. We agree with the reviewer that prior methods have achieved good performance with a single state-only expert trajectory. Our intention was not to claim novelty in adopting this setting, but rather to highlight that the single-trajectory regime is of broad interest to the community and therefore represents an important problem to study.
> >
> > Our contribution lies in showing that, within the context of Wasserstein AIL, expert-level performance can be achieved by explicitly addressing the metric misalignment inherent in the Wasserstein dual formulation. This aspect is specific to the Wasserstein AIL framework and, to our knowledge, had not been resolved in prior work. We believe this offers meaningful insight for the line of research on Wasserstein-based learning, where the choice of distance metric plays a central role.
> >
> > **Q2. Clarification on ICVF vs. AIfO.**
> >
> > We would like to clarify that we did not intend to compare ICVF (an embedding method) directly with GAIfO (an IL algorithm) as functional equivalents. Rather, we intended to contrast how different IL frameworks utilize the specific "low-quality, action-free" offline dataset available in our setting. We clarify that the comparison concerns **what data could be useful**: Standard AIfO (e.g., GAIfO) can only make use of expert data (either state or state-action) and **state-action non-expert data**; in contrast, ICVF makes **action-free non-expert data** useful.

---

> > > ### Author Response · Authors · 2025-11-26
> > > **Response to Reviewer EztW's Follow-up (2/2)**
> > >
> > > **Q3. Is such an embedding not strictly necessary for other AIL formulations, such as those based on a BCE loss (e.g., GAIfO or DACfO)?**
> > >
> > > Yes, because these methods are optimizing objectives based on $f$-divergence (e.g. KL, $\chi^2$, JS) that are not geometric-aware. Adding ICVF embedding to these methods could potentially help, but it is not strictly necessary. We next discuss the limitations of relying on $f$-divergence in this context.
> > >
> > > **Q4. Why should one prefer the Wasserstein distance in this context?**
> > >
> > > We thank the reviewer for the advice; we agree that the rationale for using Wasserstein distance should be discussed in more detail. We plan to add the following discussion to the paper:
> > >
> > >  “Adversarial Imitation Learning (AIL) methods, which learn by matching the distribution of agent states to that of the expert, are a popular approach to LfO.”  (line 40-42)
> > >
> > > Many such AIL methods measure the distribution difference by $f$-divergence, such as KL-, $\chi^2$- or JS-divergence. However, such measures have two shortcomings:
> > >
> > > 1) $f$-divergences are unaware of the underlying geometric of the states. For example, consider two agents in a gridworld aiming for the top-right corner of the grid; agent A lingers on the adjacent grid of the goal, while agent B lingers on the bottom-left corner. Measured by $f$-divergence, both agents are almost the same bad (unless agent A shares other states on the path with the expert, which is not necessary the case, especially when the trajectory is subsampled - see Appendix E.2.4 for experiments); however, with a Manhattan distance as the underlying metric, Wasserstein-based objective clearly gives more informative signal for learning.
> > >
> > > 2) $f$-divergences requires “distribution coverage”, i.e., the distributions in $f$-divergence must be on the same support set to avoid numerical error. This could lead to additional theoretical constraints, especially when an additional dataset of non-expert data is involved; for example, offline LfO methods such as SMODICE [2] require the non-expert state distribution to fully cover the expert state distribution, which does not necessarily hold in practice especially when the non-expert data quality is low (e.g. random).
> > >
> > > To address these limitations, (line 43) “the Wasserstein distance has gained popularity (Arjovsky et al., 2017; Zhang et al., 2020b). However…”
> > >
> > > We will add these discussions in the revision, and we hope our response can address the reviewer’s remaining concern. We thank the reviewer again for the time devoted into reviewing our paper, and we are happy to discuss more if the reviewer has further questions.
> > >
> > > **References**
> > >
> > > [1] S. Yue et al. Ollie: Imitation learning from offline pretraining to online finetuning. In ICML, 2024.
> > >
> > > [2] Y. J. Ma et al. Smodice: Versatile offline imitation learning via state occupancy matching. In ICML, 2022.

---

> > > > ### Comment · Reviewer_EztW · 2025-11-27
> > > >
> > > > Thank you for your response. I do not have further follow-up questions or remarks. While I appreciate the clarifications, my overall assessment remains largely unchanged. In my view, the contribution is still relatively narrow, as it primarily applies to the Wasserstein formulation of AIL, and the paper would benefit from a stronger motivation for focusing on this setting. That said, there is some value in the analysis of ICVF within this context.
> > > >
> > > > Overall, I do not believe the paper should be accepted in its current form, although I would not strongly oppose acceptance if the other reviewers and the area chair judge the contribution to be sufficient.

---

> > > > > ### Author Response · Authors · 2025-12-03
> > > > >
> > > > > We sincerely thank the reviewer for the continued engagement, the decision to raise the score and for acknowledging the value of our ICVF analysis and the new navigation experiments.
> > > > >
> > > > > However, regarding the concern that the contribution is "relatively narrow" because it focuses on Wasserstein AIL, we respectfully present a counter-perspective of the broad implications of this work.
> > > > >
> > > > > **1. Wasserstein AIL is a major, fundamental research direction.**
> > > > >
> > > > > While our work focuses on the Wasserstein formulation, this is not a narrow sub-case as Wasserstein AIL addresses fundamental limitations of $f$-divergences. This theoretical advantage has driven a sustained stream of research in top venues as we listed in the related work section (e.g., PWIL at ICLR 2021, IQ-Learn at NeurIPS 2021, and PW-DICE at ICML 2024). Thus, solving the core weakness of this framework – the Euclidean metric misalignment – is not a minor tweak. It is a fundamental fix to the underlying assumption of the entire Wasserstein IL class. By providing a principled solution to this problem, our LWAIL unlocks the true potential of Wasserstein AIL, which is of broad interest to the extensive community working on imitation learning.
> > > > >
> > > > > **2. The "Current Form" has been significantly strengthened.**
> > > > >
> > > > > Following the reviewer’s advice, we again revised the manuscript (changes marked in purple), which emphasizes both the importance of Wasserstein distance in IL/AIL and our focus on this setting. The paper is now significantly strengthened by all the reviewers’ constructive feedback; with these changes, we believe the updated version is now ready for publication in its current form.

---

### Official Review · Reviewer_XUbE · 2025-10-31

**Soundness:** 2
**Presentation:** 3
**Contribution:** 3
**Rating:** 6
**Confidence:** 4

**Summary:**

The paper proposes a state-only imitation learning approach that combines a Wasserstein Adversarial Imitation Learning objective with a learned latent representation (ICVF). The motivation is that Wasserstein distances in the original state space can behave poorly for high-dimensional or redundant dynamics, and that learning an approximately linear latent embedding can yield a geometry where Euclidean distances better reflect transition similarity. The authors provide empirical results on MuJoCo control tasks, demonstrating that training in such a latent space improves sample efficiency and stability in low-demonstration regimes.

**Strengths:**

-The idea of combining a latent, dynamics-aware embedding with a Wasserstein AIL objective is intuitive and well-motivated.
-The results show consistent gains over baselines in low-data settings, and the ablations suggest that learning a latent embedding indeed facilitates imitation learning.
-The paper provides a wide range of ablations (different embeddings, action noise, dynamics mismatch), which adds credibility to the empirical findings.

**Weaknesses:**

-Overstated novelty. Methodologically, the approach is largely a combination of known components (Wasserstein AIL and ICVF-based embeddings) rather than a fundamentally new algorithm. The paper currently presents it as a new framework rather than as a targeted study of how these parts interact.
- Fairness of comparisons. While the paper shows LWAIL with and without the latent, it does not systematically test whether other IL algorithms (e.g., IQ-Learn, OPOLO, GAIfO) would also benefit from the same latent embedding. While they have a short comparison in the appendix details about the comparison are missing (e.g. how where the hyperparameters selected). This makes it unclear whether the improvement comes from the Wasserstein loss or simply from using ICVF features with the other improvements like the adapted reward.
- Experimental focus and clarity. Important comparisons are buried in the appendix, and it is difficult to attribute which component (embedding, loss type, or regularization) drives the gains. The main tables mostly cover simple MuJoCo tasks, with only a limited exploration of harder or higher-dimensional environments.
- Limited evaluation breadth. The experiments remain confined to low-dimensional, low-contact locomotion tasks (Hopper, Walker2D, HalfCheetah, Ant). The approach’s claimed robustness would be more convincing with higher-dimensional or vision-based control tasks. For example why was the humanoid environment left out?
- Presentation imbalance. Some theoretical and background sections could be shortened to leave space for clearer experimental structure and discussion of assumptions. Key ablations and tuning details should be surfaced from the appendix.

**Questions:**

To make the contribution clearer and more convincing, I would recommend the following revisions before acceptance:

- Clarify contribution framing. Recast the paper as an investigation of Wasserstein AIL in learned latent metricsrather than as a new algorithm.
- Systematic baseline parity. Add a study where the same ICVF embedding is used across strong baselines (IQ-Learn, GAIfO, OPOLO) under searched hyperparameters and matched general setups. This would clarify whether the Wasserstein component itself is crucial. Describe how hyperparameters for baselines and variants were chosen, especially when embedding features are shared.
- Why was the humanoid environment left out? Why is there not at least one more complex or high-dimensional task (e.g., Humanoid or a visual D4RL environment). A test for scalability and the generality of the embedding is missing. Is the claim that the method can learn from on expert trajectory legitimate when only the “simple” mujoco tasks where tested?
- Highlight experimental design. Why are the key ablations (embedding vs. no-embedding, Wasserstein vs. f-divergence, baselines with latent embeddings) not all in the main paper? Why are there no clear performance summaries and standard deviations over multiple seeds for the ablations?
- Theory scope and assumptions. Discuss more explicitly when the linear-in-embedding assumption of Theorem 3.1 holds or fails. Does it fail in a truly stochastic environment?

---

> ### Author Response · Authors · 2025-11-24
> **Rebuttal to Reviewer XUbE (1/3)**
>
> Thanks for your review and constructive feedback. Below are our responses:
>
> **Q1. Overstated novelty.** (Question 1, Weakness 1)
>
> We thank the reviewer for the thoughtful comment. While our method indeed builds on existing components (Wasserstein AIL and ICVF embeddings), our contribution is *not a simple combination of known parts*. Rather, the core contribution lies in *a key insight about their interaction* that addresses a fundamental limitation in prior Wasserstein AIL methods.
>
> 1. **Our contribution goes beyond integrating known methods.**
>
> A central insight of our work is that ICVF embeddings induce a geometry in which Euclidean distance meaningfully reflects dynamical similarity between states. This directly resolves the core challenge in prior Wasserstein AIL using KR duality, where the Euclidean distance in raw state space is misaligned with dynamics. This issue has been discussed in prior works [1,2], but, to the best of our knowledge, no prior AIL method has introduced a principled solution within the dual Wasserstein formulation. PW-DICE [2] and PWIL [3] avoid this weakness by switching to primal formulations and adding additional surrogates, but do not solve the geometry-alignment issue. Thus, *our contribution is the first to introduce a remedy for this known limitation, leading to a cleaner, more principled Wasserstein AIL objective*.
>
> 2. **Our setting and objective are novel and technically meaningful.**
>
> Beyond the insight above, our work also departs from prior AIL literature in important ways:
>
> - Our embedding targets a highly challenging and underexplored regime: *only 10K transitions of low-quality, observation-only data*, potentially from different embodiments and action spaces, whereas prior work typically assumes 100K-1M *state-action* pairs.
>
> - Our formulation minimizes *state-pair transition distributions* within Wasserstein AIL, which, to our knowledge, has not been done before.
>
> - As noted by Reviewer mxsi, our method achieves *expert-level performance using a single state-only expert trajectory* in locomotion tasks, underscoring both the difficulty and practical importance of our setting.
>
> These aspects reflect substantive novelty in both problem formulation and technical perspective, even though the components are individually known.
>
> 3. **Our original submission did not claim a fundamentally new framework.**
>
> We appreciate the opportunity to clarify this point. Throughout the paper, we explicitly presented our method as an algorithm within the Wasserstein AIL family, not as an entirely new framework. For example in the original submission:
>
> - Line 79-80: “…this new latent space as the cost function *within a standard Wasserstein AIL framework*.”
>
> - Line 88-90: “…for state-based Wasserstein occupancy matching.”
>
> - Line 441-442: “our objective … is a special case of IQ-learn with Wasserstein distance,” followed by discussion of differences.
>
> - Line 809-811: “our work belongs to … the Wasserstein distance as the measure between occupancies.”
>
> To avoid any possible misunderstanding, we will further refine several phrases:
>
> - Line 74: “a novel AIL framework” -> “a novel Wasserstein AIL algorithm.”
>
> - Line 90-91: “a simple but effective method” -> “a simple but effective improvement to current Wasserstein AIL methods.”
>
> We are grateful for the reviewer’s feedback, and we welcome guidance on any additional sentences that appear overstated so that we can adjust them accordingly.
>
> **Q2. Experimental clarity.** (Question 3)
>
> Thanks for pointing that out. We have moved several important ablation studies to the main paper for better clarity and balance; see our updated PDF. We would appreciate further suggestions on which parts could be improved or emphasized for clearer presentation.
>
> Regarding which component drives the gains, we have clarified the contribution of embedding in the ablation study presented in the main paper, where we evaluate how well our dynamic-aware embedding works. As for the loss type and regularization, we assume the reviewer is referring to the training details of the reward network used in the Wasserstein discriminator. We agree that these design choices are important; however, they follow standard and widely adopted practices in Wasserstein-based learning, and are not unique design decisions of our method.

---

> ### Author Response · Authors · 2025-11-24
> **Rebuttal to Reviewer XUbE (2/3)**
>
> **Q3. Fair comparison and baseline parity.** (Question 2, Weakness 2)
>
> 1. We appreciate the reviewer’s suggestion to isolate the sources of performance gain. We have explicitly investigated the effect of transferring the ICVF embedding to other baselines (IQ-Learn [4], WDAIL [5], GAIL). As presented in Tab. 4, utilizing the ICVF embedding within these baselines did not yield improvements compared to our proposed method. This result reinforces our core insight regarding the role of ICVF within the Wasserstein AIL framework, as clarified in the novelty discussion above. Note, OPOLO is not compatible with ICVF, as it relies on learning an inverse dynamics model $p(a|s,s’)$; however, the ICVF embedding $\phi(\cdot)$ is not an injective function, and thus it will introduce unnecessary noise in modeling $p(a|\phi(s),\phi(s’))$ by mixing different distributions of $a$ from different $(s,s’)$. And GAIfO is basically the LfO version of GAIL.
>
> 2. To ensure a fair and reproducible comparison, we adopted the hyperparameter configurations reported in the original implementations of all baselines. We made this clear in our original Appendix D.4, line 1011 (now 1130), “we use default settings of those repos, except for the number of expert trajectories and the subsampling factor.” We also deliberately avoided extensive hyperparameter tuning for our own method to prevent overfitting to specific tasks (please refer to our response to Reviewer mxsi, Q2, for experiments about hyperparameters). Given that we used standard settings for baselines and a fixed configuration for our method, we believe that the significant margin of improvement reflects the robustness of the proposed approach rather than hyperparameter sensitivity.
>
> 3. To further address the reviewer’s concern, we tuned the hyperparameters of the baseline methods, IQ-learn and WDAIL, as these are the two methods that we already reported results with and without ICVF in the current Tab. 6  and Tab. 1, respectively (we did not consider GAIL, as its performance is weaker than the other baselines).
>
> For IQ-learn, we tuned the following hyperparameters: learning rate {1/10x, 1/3x, 1x}; loss type {initial state value loss, all state value loss}; divergence used {chi square, Wasserstein}.
>
> For WDAIL, we tuned the following hyperparameters: learning rate {1/3x, 1x, 3x}; PPO update frequency in steps {1024, 2048, 4096}; update epochs {5, 10, 20}.
>
>
> For WDAIL, the results both with and without ICVF do not show any noticeable improvement with hyperparameter search. For IQ-Learn, we witness some performance improvement compared to results in the current Tab. 6, reported as follows (as the other methods do not conduct hyperparameter search, for a fair comparison, we do not update them in Tab. 6).
>
> Env | IQ-Learn | IQ-Learn + ICVF
> ---|---|---
> Hopper | 86.24 $\pm$ 21.92 | 71.51 $\pm$ 11.27
> HalfCheetah | 2.66 $\pm$ 3.59 | 3.82 $\pm$ 0.98
> Walker2d | 17.65 $\pm$ 7.97 |  16.49 $\pm$ 7.84
> Ant | 31.20 $\pm$ 0.82 | 27.97 $\pm$ 12.18
>
> **Q4.  Limited evaluation breadth.** (Question 4, Weakness 3)
>
> 1. We appreciate the reviewer’s concern regarding evaluation breadth. We focused on the standard MuJoCo locomotion environments (Hopper, Walker2D, HalfCheetah, Ant), because they are canonical benchmarks widely used in prior imitation learning and offline RL literature, while Humanoid is not tested by many of our direct baselines such as WDAIL and OPOLO.
> Using these environments ensures that the expert datasets are well aligned with previous work, enabling fair, directly comparable evaluations and minimizing confounding factors unrelated to the algorithmic contribution.
>
> 2. To further address the reviewer’s request for more complex or higher-dimensional tasks, we have included additional experiments in the paper on more challenging environments. All scores are normalized to 0-100 by dividing the reward with the expert trajectory’s reward  (Maze2d are normalized by D4RL settings); the higher is better.
>
> **Humanoid-v5**
>  Method        | Score
> ---|---
> LWAIL         | 24.36 ($\pm$ 2.53)
> LWAIL without  ICVF  | 7.13 ($\pm$ 0.67)
>
> **AntMaze-Umaze-v2**
> Method        | Score
> ---|---
> LWAIL         | 34.76 ($\pm$ 7.21)
> LWAIL without ICVF | 0.03 ($\pm$ 0.01)
>
> **DMControl: Ball-in-Cup**
>  Method        | Score
> --|---
>  LWAIL         | 84.85 ($\pm$ 3.55)
> LWAIL without ICVF | 14.80 ($\pm$ 4.52)
>
> We also conduct experiments on the maze2d-medium and maze2d-large in representation of navigation tasks, and add different stochasticity (noise level) on the initial state. Below is the result:
>
> **maze2d-medium**
> Noise Level | LWAIL            | Without LWAIL
> ---|---|---
>  0.0         | 144.59 ($\pm$ 0.87)          | 117.91 ($\pm$ 0.30)
>  0.2         | 143.42 ($\pm$ 0.91)          | -17.00 ($\pm$ 0.01)
>  0.5         | 135.12 ($\pm$ 3.12)          | -16.98 ($\pm$ 0.02)
>
> (Continued in next part)

---

> ### Author Response · Authors · 2025-11-24
> **Rebuttal to Reviewer XUbE (3/3)**
>
> (Continued from last part)
>
> **maze2d-large**
> Noise Level | LWAIL            | Without LWAIL
> ---|---|---
>  0.0         | 156.68 ($\pm$ 0.60)          | 148.25($\pm$ 6.41)
>  0.2         | 157.04 ($\pm$ 0.71)          | -11.11 ($\pm$ 0.04)
>  0.5         | 157.06 ($\pm$ 0.81)          | -11.13 ($\pm$ 0.02)
>
> These results demonstrate that our method remains robust and effective in higher-dimensional, long-horizon, and more complex control settings, supporting the generality of our approach beyond low-dimensional locomotion.
>
> **Q5. Highlight experimental design and presentation imbalance.** (Question 5, Weakness 4)
>
> Thanks for pointing out; we have modified the PDF accordingly and put more experiments into the main paper. Note, we reported standard deviations for every table presented in our original submission, and stated in original line 356 (now 358) that “We report mean and standard deviation from 5 independent runs with different seeds.” We also stated in Tab. 1 that “Results are averaged over 50 trajectories.”
>
> **Q6. Theory scope and assumptions.** (Question 5)
>
> The linear-in-embedding assumption of Thm. 3.1 holds for any **optimal policy** $\pi_z$ with goal $z$, when:
>
> - The agent takes the shortest path towards its goal state before reaching it, and
>
> - After reaching the goal state, the agent stays at the goal state or keeps returning to it as soon as the environment allows (e.g., keeping a pendulum inverted after reaching its balance).
>
> It fails when the agent often takes a further path to move between the dynamically adjacent state, such that the second term in the last line (which is the probability of an optimal policy taking a longer path due to environment noise) of Eq. 14 in case one is non-negligible. If the environment is fully deterministic, then this term becomes 0, and the approximations in Thm. 3.1 become strict equations. For case two, it does not matter if the environment is non-negligible; as long as the environment is consistent with respect to timestep and the agent keeps returning to the goal state as soon as possible, $\sum_{i=2}^{\infty}\gamma^{i-1}p(s_{i+t}=s_+|s_t=s)$ will be a multiple of $p(s_{t+i}=s_+|s_t=s)$ and the conclusion still holds.
>
> Additionally, to simulate a stochastic environment, we inject Gaussian noise with a standard deviation of 0.1, 0.2 and 0.5 into the actions performed into the MuJoCo environments (note that 0.5 is a very large noise given that the actions are in [-1, 1]). This noise introduces controlled randomness, emulating real-world uncertainty and variability in system responses. The experiment setting is following the main experiment, and here are the results:
>
> Env | Hopper | HalfCheetah | Walker | Ant | Average
> ---|---|---|---|---|---
>  LWAIL | 110.52 ($\pm$1.06) | 86.71 ($\pm$5.67) | 105.30 ($\pm$2.33) | 80.56 ($\pm$13.09) | 95.77
>  LWAIL with noise 0.1 | 110.25 ($\pm$1.78) | 82.04 ($\pm$6.68) | 104.98 ($\pm$1.44) | 79.95 ($\pm$12.08) | 94.31
>  LWAIL with noise 0.2 | 108.27 ($\pm$2.38) | 79.29 ($\pm$5.21) | 104.61 ($\pm$2.34) | 77.10($\pm$12.08) | 92.32
>  LWAIL with noise 0.5 | 107.93 ($\pm$2.38) | 48.39 ($\pm$3.95) | 41.28 ($\pm$29.66) | -12.94($\pm$9.12) | 46.17
>
> The results clearly show that our method is robust to stochastic environments.
>
> **Reference**
>
> [1] J. Stanczuk et al. Wasserstein GANs Work Because They Fail (to Approximate the Wasserstein Distance). arXiv, 2021.
>
> [2] K. Yan et al. Offline imitation from observation via primal wasserstein state occupancy matching. In ICML, 2024.
>
> [3] R. Dadashi et al. Primal wasserstein imitation learning. In ICLR, 2021.
>
> [4] D. Garg et al. Iq-learn: Inverse soft-q learning for imitation. In NeurIPS, 2021.
>
> [5] M. Zhang et al. Wasserstein distance guided adversarial imitation learning with reward shape exploration. In DDCLS, 2020.

---

### Author Response · Authors · 2025-11-24
**Global Response**

We would like to thank the reviewers, AC, and SAC for their time devoted to reviewing our work and helping strengthen the paper.
We are encouraged that all reviewers found the paper clearly written, well-structured, and easy to follow. Importantly, the reviewers highlight the strengths of our contributions:

1. **Reviewer XUbE** recognizes that integrating a dynamics-aware latent embedding with a Wasserstein AIL objective is an intuitive and well-motivated idea, and Reviewer mxsi points out that LWAIL is novel in its integration of an ICVF-learned latent space into a Wasserstein AIL framework to create a dynamics-aware distance metric.
2. The breadth and depth of our ablation studies were repeatedly praised by **Reviewers XUbE, mxsi, and Cuqz**.

We have addressed all reviewers’ questions in our rebuttal and updated our PDF accordingly. More specifically, we made the following updates:

1. We modified the introduction to better show the connection between our work and Wasserstein AIL framework. **(Reviewer XUbE)**
2. We added expert performance in the experiment section for reference. **(Reviewer EztW)**
3. We added subsection 4.3, “Navigation Environment,” which includes experiment results on antmaze and maze2d with larger mazes as more complex environments. We also updated the environment details in the appendix accordingly. **(Reviewers XUbE, EztW, mxsi)**.
4. We simplified Fig. 4 in the main paper and deferred full results with all baselines to Appendix. **(Reviewer mxsi)**
5. We surfaced more ablations into the main paper. **(Reviewer XUbE)**
6. We added discussion on model-based imitation learning in “Extended Related Work” section in Appendix. **(Reviewer Cuqz)**
7. We added hyperparameter sensitivity analysis in Appendix. **(Reviewer mxsi)**

---

### Meta-Review · Area_Chair_M9As · 2026-01-06

**Summary:**

This paper is marginally above the acceptance threshold because, despite some concerns raised by R2 regarding novelty and limited evaluation, the authors provide sufficient rebuttals addressing substantive technical questions. R1, R3, and R4 acknowledge that the proposed method demonstrates meaningful improvements in challenging, low-data, observation-only imitation learning settings, and the experimental validation further supports the robustness and generality. The insight of using dynamics-aware embeddings within a Wasserstein AIL framework is technically motivated, with performance improvements over strong baselines. While the novelty is incremental in terms of components, the combination and analysis are sufficiently justified, and the work contributes both a practically useful algorithm and insights into embedding design for imitation from observation. Overall, the paper is slightly above the bar for acceptance.

**Reviewer Concerns:**

1. Limited novelty/contribution (R1-R4)
2. Insufficient justification for dynamics-aware latent embeddings (R1-R4)
3. Lack of systematic and fair baseline comparisons (R1, R2, R4)
4. Experimental scope and generality are limited (e.g., weak ablations and unclear attribution of gains) (R1, R3, R4)
5. Presentation and clarity issues (R1, R2, R3)

See further elaborations below (Reviewer Scores).

**Reviewer Scores:**

R4 was satisfied with the responses prior to the termination of the discussion period. R4 has transitioned from a skeptical gatekeeper to an engaged domain expert, and the rating was raised from 4 to 6. Despite an engaged discussion, R2 still found the contribution relatively narrow, as the work is primarily applied to the Wasserstein formulation of AIL, and the paper would benefit from a stronger motivation for focusing on this setting. As a result, R2 does not believe the paper should be accepted in its current form (keeping the rating as 4), although R2 would not strongly oppose acceptance

R1’s concerns are largely resolved by (i) reframing the contribution as a principled improvement within Wasserstein AIL rather than a new framework; (ii) establishing fair baseline parity, including transferring the same ICVF embedding to strong baselines (e.g., IQ-Learn, WDAIL) with hyperparameter tuning, and (iii) strengthening the experiments, moving key ablations into the main paper and adding harder, higher-dimensional tasks (Humanoid, AntMaze, DMControl, stochastic settings), thereby improving evaluation breadth and clarity. The AC feels R1 would raise or at least maintain the positive rating (6 or above).

As for R3, his/her concerns are addressed by adding experiments on higher-dimensional and long-horizon tasks (Humanoid, AntMaze, DMControl, maze2d) showing LWAIL’s robustness and generalization, providing hyperparameter sensitivity analyses demonstrating stable performance across reasonable ranges, clarifying why dynamics-aware embeddings improve learning and stability in locomotion tasks, and improving figure clarity and presentation for easier comparison with baselines. The AC also believes R3 would raise or at least maintain the positive rating (6 or above).

---

### Decision · Program_Chairs · 2026-01-26

Accept (Poster)